# Skill Learning via Policy Diversity Yields Identifiable Representations for Reinforcement Learning

**Patrik Reizinger**[*1,2], **Bálint Mucsányi**[*3,4], **Siyuan Guo**[*1,5], **Benjamin Eysenbach**[6],
**Bernhard Schölkopf**[†1,2], **and Wieland Brendel**[†1,2]

[1]Max Planck Institute for Intelligent Systems, Tübingen, Germany
[2]ELLIS Institute Tübingen, Tübingen, Germany
[3]University of Tübingen, Tübingen, Germany
[4]Tübingen AI Center, Tübingen, Germany
[5]University of Cambridge, Cambridge, United Kingdom
[6]Princeton University, Department of Computer Science, Princeton, United States

## Abstract

Self-supervised feature learning and pretraining methods in reinforcement learning (RL) often rely on information-theoretic principles, termed mutual information skill learning (MISL). These methods aim to learn a representation of the environment while also incentivizing exploration thereof. However, the role of the representation and mutual information parametrization in MISL is not yet well understood theoretically. Our work investigates MISL through the lens of identifiable representation learning by focusing on the Contrastive Successor Features (CSF) method. We prove that CSF can provably recover the environment's ground-truth features up to a linear transformation due to the inner product parametrization of the features and skill diversity in a discriminative sense. This first identifiability guarantee for representation learning in RL also helps explain the implications of different mutual information objectives and the downsides of entropy regularizers. We empirically validate our claims in MuJoCo and DeepMind Control, and show that CSF provably recovers the ground-truth features from both states and pixels. Our code is available at `https://github.com/bmucsanyi/identifiable-misl`.

## 1 Introduction

The field of *Reinforcement Learning (RL)* faces several challenges, such as learning under sparse rewards, exploring the environment, and designing an appropriate reward function. Many solutions use different self-supervised approaches including curiosity, intrinsic motivation, or unsupervised skill discovery (USD) (Eysenbach et al., 2019; Sharma et al., 2020; Pathak et al., 2017; Pathak et al.; Sancaktar et al., 2023; Ha & Schmidhuber, 2018). mutual information skill learning (MISL) methods are a subclass of USD that use Mutual Information (MI) objectives. Despite this common design factor, they exhibit wildly varying performance (Park et al., 2024b; Zheng et al., 2025), which is not yet well understood theoretically. Empirical evidence suggests guidelines on, e.g., parametrizing the action-value function (Zheng et al., 2025; Liu et al., 2024), but it does not explain how similar principles lead to such large differences in performance.

We build upon recent theoretical advancements in self-supervised learning (SSL), particularly *nonlinear Independent Component Analysis (ICA)* theory (Zimmermann et al., 2021; Hyvarinen et al., 2019; Reizinger et al., 2024a; Roeder et al., 2020) and *Causal Representation Learning (CRL)* (Schölkopf et al., 2021; Wendong et al., 2023; Reizinger et al., 2024b; Rajendran et al., 2023; Guo et al., 2023). Identifiability results derive guarantees on learning the latent factors from data, i.e., recovering the underlying data generating process (DGP) from high-level observations such as pixels. Identifiability results are relevant for RL in the partially observable Markov Decision Processes (POMDPs), as the states are not always observed. Thus, the goal is to provably infer the ground-truth states from

---

[*]Equal contribution. Correspondence to `patrik.reizinger@tuebingen.mpg.de`.
[†]Equal supervision.

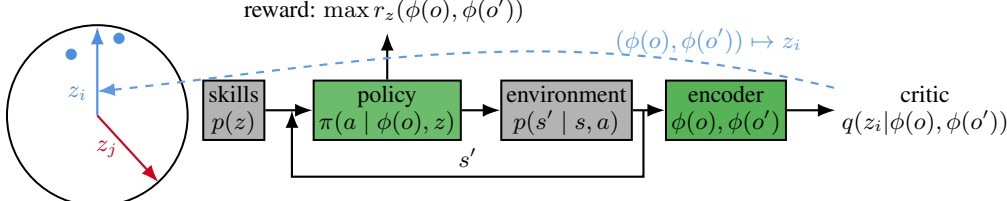

Figure 1: **The building blocks of mutual information skill learning (MISL) (Zheng et al., 2025)**: The MISL method Contrastive Successor Features (CSF) (Zheng et al., 2025) uses uniformly drawn skills on the hypersphere to learn a skill-conditioned policy by maximizing skill diversity via Eq. (3). CSF learns an encoder $\phi$ to map observations $o$ to features $\phi(o)$, and uses an inner product parametrization in the critic $q(z_i|\phi(o), \phi(o'))$, where $q$ is trained via a contrastive loss (1) to infer the skill $z_i$ from features of consecutive states. Intuitively, this is only possible if each skill is representative of only a subset of states. The skill distribution and the environment dynamics are fixed, whereas the policy and the encoder are learned, for details, refer to § 4. **Main result:** We formalize the underlying assumptions of CSF with a probabilistic DGP on the hypersphere, then, building on prior works from identifiability, prove that the building blocks of CSF are such that after training, the learned features identify the ground-truth states up to a linear transformation, which explains practical differences between some objectives and parametrizations (§ 4.2).

observations. Intuitively, to achieve identifiability, the training samples should cover all aspects of the underlying world model. That is, they require diversity (also called sufficient variability), mirroring the goal of USD to explore and learn diverse skills. In the same way that identifiability theory helps explain the success of self-supervised pretraining in computer vision (Zimmermann et al., 2021; Rusak et al., 2025; Ibrahim et al., 2024; Reizinger et al., 2024a), we use recent insights from identifiability to study self-supervised RL methods. Using Zheng et al. (2025) as a prototypical MISL method, we provide theory to elucidate why and when MISL works by proving that Contrastive Successor Features (CSF) identifies the ground-truth states of the underlying Markov Decision Process (MDP) (§ 4.1). Our analysis also helps reason about failure cases and formulate practical recommendations (§ 4.2).

Thus, we focus on identifying the components of successful MISL methods, following Reizinger et al. (2024a); Park et al. (2023a). We show that

*The good performance of MISL hinges on learning representations via mutual information estimation under diverse policies and an inner product parametrization of the model.*

Identifiability also sheds light on the limitations of some methods and design choices (§ 4.2), e.g., why the maximum-entropy policy is suboptimal in skill learning, or why feature parametrization matters. By drawing on diversity and variability assumptions from the ICA and CRL literatures, we formalize what constitutes a diverse policy and analyze the roles of feature dimensionality and skill space coverage, yielding practical insights. Our **contributions** are (Fig. 1):

- We explain the success of mutual information skill learning (MISL) as the interplay of MI estimation under diverse policies and inner product model parametrization, leading to the first identifiability results in RL for Contrastive Successor Features (CSF) (§ 4.1);
- Our theoretical results enable us to quantify what a diverse policy means and to pinpoint limitations of previous methods, leading to practical recommendations (§ 4.2);
- We validate our theoretical claims of feature identifiability in CSF in both state- and pixel-based MuJoCo and DeepMind Control environments (§ 5).

## 2 RELATED WORK

### 2.1 UNSUPERVISED SKILL DISCOVERY

While reinforcement learning (RL) is typically cast as a problem of learning a single policy to maximize a scalar reward function (Sutton et al., 1998), recent research focused on self-supervised objectives, inspired by information theory, to incentivize exploration while side-stepping the problem of reward specification (Park et al., 2024b;a; Eysenbach et al., 2019; Sharma et al., 2020; Choi et al., 2021). Such methods aim to learn "universal" representations that can be used to solve diverse downstream tasks. Paradigms include curiosity (Pathak et al., 2017; Burda et al.), regularity (Sancaktar et al., 2023), model uncertainty (Treven et al., 2023), and disagreement (Pathak et al.; Sekar et al.,

2020; Mendonca et al., 2023), with works showing connections to Contrastive Learning (CL) (Eysenbach et al., 2022; 2023; Zheng et al., 2023; Hansen et al., 2019; Park & Pardalos, 2021; Laskin et al., 2022). Other works in *unsupervised skill discovery (USD)* aim to learn a set of policies (i.e., "skills") that span the space of behaviors that an agent might perform in an MDP (Eysenbach et al., 2019; Achiam et al., 2018; Zheng et al., 2025; Sharma et al., 2020; Park et al., 2022; Sutton et al., 1998). *Mutual information skill learning (MISL)* is a subset of USD methods that relies on information-theoretic principles (Zheng et al., 2025; Park et al., 2024b; Sharma et al., 2020; Eysenbach et al., 2019; Mohamed & Jimenez Rezende, 2015; Gregor et al., 2017; Achiam et al., 2018). Yang et al. (2025) investigate MISL from a skill learning perspective, though they define the skill to include the parameters of the policy network.

## 2.2 Self-Supervised Learning

Self-supervised methods aim to use a pretraining task to learn "universal" representations that facilitate solving downstream tasks, such as classification, object detection, or segmentation (Chen et al., 2020; Oquab et al., 2024; Radford et al., 2021; Bardes et al., 2022; Balestriero et al., 2023; Balestriero & LeCun, 2022; Siméoni et al., 2025). Many SSL methods are related to information-theoretic principles (Zimmermann et al., 2021; Bizeul et al., 2024; Liu et al., 2022; Shwartz-Ziv et al., 2022). These methods often aim to learn representations such that similar samples have similar representations and *dis*similar samples *dis*similar representations (Wang & Isola, 2020). Recent advancements in the nonlinear ICA literature explained the success of many contrastive SSL algorithms by proving their identifiability (Zimmermann et al., 2021; Rusak et al., 2025; Reizinger et al., 2024a).

## 2.3 Causal Representation Learning

Recent work has shown fundamental connections between RL, SSL, and causality. For example, there is a long line of work using self-supervised learning to drive RL, particularly in skill discovery (Eysenbach et al., 2019; 2022; Sharma et al., 2020; Pathak et al., 2017; Pathak et al.; Hansen et al., 2019; Park et al., 2024b; 2023b). Indeed, the RL problem is fundamentally about agents taking interventions (i.e., actions), pinpointing the natural connection to the field of causality (Pearl, 2009; Spirtes et al., 2000)—which is to answer causal queries about the effect of interventions. That is, to predict what happens when an agent takes a series of actions. Thus, naturally, many CRL methods are often tested on tasks involving an agent such as a robotic arm or a control system (Schölkopf et al., 2021; Liu et al., 2023; Lippe et al., 2022a;b; Yang et al., 2023).

# 3 Background

**Notation.** We consider a partially observable Markov Decision Process (POMDP) without a reward function. We denote successive states as $s$ and $s'$ with initial state distribution $p(s)$, actions as $a$, the state transition distribution as $p(s' \mid s, a)$, observed variables as $o = g(s), g \colon \mathbb{R}^d \to \mathbb{R}^{D \geq d}$ being a deterministic generator function, representations or features as $\phi(o)$, and skills as $z_i$, which are all random variables (RVs) sampled from a prior skill distribution $p(z)$. The skill-conditioned policy is $\pi(a \mid o, z)$, and the variational model for the critic is $q(z \mid \phi(o), \phi(o'))$.

## 3.1 Identifiability in Self-Supervised Learning.

*Identifiability* means that, assuming an underlying DGP for the data, the corresponding "ground-truth" latent factors can be recovered up to simple transformations (such as linear maps). That is, for ground-truth states $s$ and features $\phi(o)$, it holds for a particular linear map $\mathbf{A}$ that $\phi(o) = \mathbf{A}s$. In the nonlinear case, identifiability is only possible with further assumptions (Darmois, 1951; Hyvärinen & Pajunen, 1999; Locatello et al., 2019), which restrict either the model class (e.g., to have a specific Jacobian structure (Gresele et al., 2021)) and/or the latent distribution (e.g., nonstationary time series (Hyvarinen & Morioka, 2016)). Intuitively, these assumptions are required to "break the symmetries", e.g., the rotational symmetry of the Gaussian distribution or the rotational symmetry of the inner product parametrization. These symmetries emerge from the likelihood—most methods can be thought of as maximizing the data likelihood or a related quantity (such as cross-entropy)—, where we plug in the model parametrization and the learned latent distribution via a change of variables. ICA theory shows that estimating MI is generally insufficient to learn a "useful" representation without further assumptions (Tschannen et al., 2020; Roeder et al., 2020; Hyvärinen & Pajunen, 1999; Locatello et al., 2019; Reizinger et al., 2024a).

A prominent family is that of *auxiliary* variable methods (where the latents are conditionally independent given the auxiliary variable) (Hyvarinen et al., 2019; Gresele et al., 2020; Khemakhem et al., 2020a; Hälvä et al., 2021; Hyvarinen & Morioka, 2016; Khemakhem et al., 2020b; Locatello et al., 2020; Morioka & Hyvarinen, 2023; Morioka et al., 2021)—as we will show, skills in MISL can also be interpreted as auxiliary variables. Intuitively, diverse skills are representative of a set of *distinct* states. To prove identifiability, ICA usually assumes specific DGPs, such as energy-based models or inner product parametrization. Recently, Reizinger et al. (2024a) showed that the cross-entropy loss is key to explaining why many (self-)supervised deep learning models learn useful (identifiable) representations. We use this insight to prove the identifiability of CSF.

## 3.2 MUTUAL INFORMATION SKILL LEARNING (MISL)

**Representation learning.** MISL is based on representation learning, providing a form of information bottleneck, which is crucial for success (Zheng et al., 2025; Park et al., 2021). The learned representation captures the relationship between $(s, s')$ and $z_i$, including state transition information. The representation is usually constrained to the (unit) hypersphere, following the practice of CL (Chen et al., 2020). To reflect the dynamics of the environment, i.e., the relationship between consecutive states $(s, s')$, it is common to use the feature differences in the objective function, i.e., $\phi(o') - \phi(o)$ with encoder $\phi$. For a detailed review on representation learning in RL, cf. Echchahed & Castro (2025).

**Diverse skills: mixture policies.** MISL aims to learn distinguishable, i.e., *diverse* skills—assuming that solving a variety of tasks requires a versatile policy. This is implemented via a skill-conditioned policy, i.e., $\pi(a|o, z)$ which is trained to maximize diversity. Intuitively, the marginal policy $\pi(a|s) = \int \pi(a|o, z)p(z)$ can be thought of as a mixture policy, where each mixture component is a different tool in the agent's toolbox (cf. Ex. 1). Diversity means that a discriminative model can uniquely infer the skill $z_i$ from consecutive states $(s, s')$. This notion relates to sufficient variability conditions in ICA (Hyvarinen & Morioka, 2016; Hyvarinen et al., 2019; Khemakhem et al., 2020b) and interventional discrepancy from CRL (Wendong et al., 2023). We refer the reader to Reizinger et al. (2024b) for a discussion on different ways of measuring diversity. Formally:

**Definition 1** (Diverse skill-conditioned policies). *For a set of skills $z_i \in \mathcal{S}^{d-1}$ that form an affine generator system of $\mathbb{R}^d$ (cf. Defn. 2), we call a skill-conditioned policy $\pi(a|o, z)$ diverse if an ideal discriminative model can uniquely infer the skill from consecutive states $(s, s')$, given a set of skills. Alternatively, $\pi(a|o, z)$ is diverse, if for given state transitions $p(s'|s, a)$, the integral $\int p(s'|s, a)\pi(a|o, z_k)p(s|z_k)ds$ is not equal almost surely for any two $z_i, z_{j\neq i}$.*

**Architecture: inner product parametrization.** MISL objectives approximate MI via an evidence lower bound (ELBO), for which they require a variational approximation. The variational posterior (i.e., the critic or the $Q$-value function) is often parametrized as an inner product, which is critical for achieving great performance (Zheng et al., 2025), though an explanation is yet to be provided. This parametrization is prevalent in SSL, and was theoretically shown to be crucial for identifiability guarantees (Roeder et al., 2020; Zimmermann et al., 2021; Hyvarinen et al., 2019; Hyvarinen & Morioka, 2016; Khemakhem et al., 2020b; Reizinger et al., 2024a).

**Contrastive Successor Features (CSF) (Zheng et al., 2025).** Our identifiability proof in § 4 is for a prototypical MISL method, CSF (Zheng et al., 2025), which is representative of prior work (Park et al., 2024b; Gregor et al., 2017; Warde-Farley et al., 2018). CSF learns both a feature representation via an encoder and a skill-conditioned policy; the skills are represented by vectors $z$ drawn uniformly from the hypersphere (cf. Fig. 1).

**State representation.** CSF learns a probabilistic critic $q(z|\phi(o), \phi(o'))$ with encoder $\phi$ to discriminate the skills based on consecutive observations $(o, o')$ corresponding to the state transition $(s, s')$. The encoder can be trained either from direct state observations or from pixels. The loss is a contrastive lower bound on the mutual information $I(s, s'; z)$:

$$q(z_i|\phi(o), \phi(o')) = \frac{p(z_i)\exp\left[(\phi(o') - \phi(o))^\top z_i\right]}{\mathbb{E}_{p(z)}\exp\left[(\phi(o') - \phi(o))^\top z\right]}, \tag{1}$$

which is equivalent to a cross-entropy loss, as it was shown for different parametrizations (Hyvarinen & Morioka, 2016; Hyvarinen et al., 2019; Zimmermann et al., 2021; Rusak et al., 2025). Intuitively, contrastive losses are used to learn a probabilistic model over latent vectors that prescribes the

relationship between similar and dissimilar samples. To fit this distribution, they minimize a statistical distance (the Kullback-Leibler divergence), which relates the loss to cross entropy. The loss in (1) can be equivalently seen as a parametric instance discrimination objective (Wu et al., 2018; Oord et al., 2019; He et al., 2020) on the feature difference $(\phi(o') - \phi(o))$, akin to formulations in prior work (Ibrahim et al., 2024; Reizinger et al., 2024a). Parametrizing the critic as a log-linear model with an inner product parametrization is crucial for identifiability (Hyvarinen et al., 2019; Roeder et al., 2020; Zimmermann et al., 2021; Reizinger et al., 2024a).

**The policy.** The skill-conditioned policy $\pi(a \mid \phi(o), z)$ is learned by optimizing the reward function $r_z(\phi(o), \phi(o')) = (\phi(o') - \phi(o))^\top z$, where $\phi$ is the same encoder as above:

$$\pi = \arg\max_{\pi} \mathbb{E}_{p(z)} \mathbb{E}_{\pi(\cdot|\cdot,z)} \left[ \sum_{t=0}^{\infty} \gamma^t r_z(\phi(o), \phi(o')) \right] = \mathbb{E}_{p(z)} \left[ \mathbb{E}_{\pi(\cdot|\cdot,z)} \left[ \sum_{t=0}^{\infty} \gamma^t (\phi(o') - \phi(o)) \right]^\top z \right], \quad (2)$$

$$\text{where} \quad s' \sim p(s' \mid s, a), \quad a \sim \pi(a \mid o, z), \quad p(z) \stackrel{d}{=} \text{Uniform}(\mathcal{S}^{d-1}). \quad (3)$$

### 3.3 BUILDING THE BRIDGE: MODELING THE MISL PROBLEM AS A DGP.

Real-world RL problems often only provide high-dimensional observations $o$ but no access to the ground-truth states $s$. MISL methods learn a representation $\phi(o)$ of these observations and use those downstream to determine the next action. We investigate whether and to what extent the learned representation $\phi(o)$ captures important information about the states $s$. We use tools from identifiability theory to answer this question. For this, we need to formally define a data generating process (DGP), which connects the POMDP of the RL problem to a latent variable model. In essence, the DGP posits a probabilistic model of states, actions, transitions, and skills. Identifiability guarantees require assumptions on the DGP, which requires us to transform the design choices of MISL methods into formal assumptions. To match the ICA literature, we model the skills as a set, with each skill having a corresponding high-dimensional unit vector $z_i \in \mathcal{S}^{d-1}$—this way, the skills can be treated as the auxiliary variables in the ICA literature. This is slightly different from how MISL methods handle the skills by sampling them for each rollout from, e.g., a uniform $p(z)$. To reconcile this modeling difference, we experimentally compare a fixed set of skills (of varying number) and skill sampling in Fig. 4. However, as identifiability guarantees only require that the skills span $\mathbb{R}^d$, these modeling choices are compatible, as with enough samples, the skill vectors span $\mathbb{R}^d$ almost surely. The POMDP includes the model of the state transitions $p(s'|s, a)$, from which we observe $a$, but might not directly observe $s$, only a function thereof via the generator $o = g(s)$. The goal of ICA is to invert $g$, i.e., to extract the state $s$ from $o$. ICA further requires making assumptions about the probabilistic model, typically about a conditional distribution. As MISL methods aim to infer the skill from state pairs, we will assume a specific form for the conditional $p(s' - s|z)$, where the feature difference $s' - s$ is defined in $\mathcal{S}^{d-1}$.

## 4 IDENTIFIABILITY INSIGHTS IN MUTUAL INFORMATION SKILL LEARNING (MISL)

**Intuition.** RL and representation learning methods are often pretrained via self-supervised tasks to learn a "universal" representation that can solve many downstream tasks (§ 3.2).

*Our insight is that learning diverse skills is equivalent to learning to distinguish data under different distribution shifts or interventions, and this leads to RL agents identifying the ground-truth states of the underlying POMDP up to a linear transformation.*

Before analyzing the relevant technical assumptions, we provide an illustrative example of how distinguishing skills can be useful to learn the states of the underlying POMDP:

**Example 1.** *Assume that a robot moves around in a maze to create a map of it. However, it does not have access to other sensors but a camera. To create the map, i.e., to learn the underlying state information such as the position of the robot, the walls, or other objects, it needs to move around to collect representative images. The ICA setting assumes that we already have such images and aims to reconstruct the state. Skill-based RL solves a harder problem, as it also needs to learn a policy to explore while learning the underlying state representations. The question we answer in this work is: do those representations identify important quantities such as position and orientation?*

## 4.1 THE IDENTIFIABILITY OF CONTRASTIVE SUCCESSOR FEATURES (CSF)

Our key insight is to analyze recent advances in MISL (Eysenbach et al., 2019; Hansen et al., 2019; Sharma et al., 2020; Park et al., 2021; Eysenbach et al., 2022; Park et al., 2023b; 2024b; Zheng et al., 2025) through the lens of identifiability theory: *The success of MISL methods, including the role of diversity and importance of a linearly parametrized critic, can be explained by identifiability theory.* To show the identifiability of the CSF features, for this section, we assume that we have access to a diverse skill-conditioned policy and recall how we related the POMDP to a DGP. We proceed in the following steps:

(i) We show that given a diverse policy, the collected data (state trajectories or observations thereof) satisfy the assumptions required for identifiability in ICA.
(ii) Then we investigate what this identifiability result implies for CSF.

**Matching the assumptions of CSF to ICA.** We investigate whether the assumptions of nonlinear ICA theory match those in MISL. The assumptions below are sufficient to recover the ground-truth states up to a linear transformation, i.e., one can fit a linear map between the features learned by the model and the true ones (e.g., if one has access to them, such as in a simulator). We state our assumptions informally (the formal statement is in Appx. B.1):

**Assumption 1** (Informal). *For consecutive states $s, s' \in \mathbb{R}^d$, $(s' - s) \in \mathcal{S}^{d-1}$ and skills $z_i \in \mathcal{S}^{d-1}$*
  *(i) There is a finite set of skills on the unit hypersphere, which are diverse in the sense of Defn. 1.*
  *(ii) Conditioning on the skill, the consecutive states $s, s'$ are close to each other.*
  *(iii) Each state difference $(s' - s)$ is marginally equiprobable.*
  *(iv) The observations $o$ are generated by passing the latent state $s$ through a continuous and injective generator, where $\dim o \geq \dim s$.*
  *(v) We train an encoder with an inner product parametrization to map observations to features $\phi$ s.t. $\dim \phi \geq \dim s$, and assume that it can globally optimize the contrastive objective in Eq. (1)*

The above assumptions are sufficient for the identifiability result; however, this *does not imply* they are all necessary. Namely, many identifiability results, despite their assumptions, can be successfully applied in practical scenarios such as ecology, climate science, robotics, functional medicine, dynamical systems, and neuroscience (Zhu et al., 2025; Wismüller et al., 2022; Locatello et al., 2020; Lippe et al., 2022b; Zhou & Wei, 2020; Yao et al., 2024b;a). Many works show that identifiability is robust to assumption violations in practice (Zimmermann et al., 2021; Sliwa et al., 2022; von Kügelgen et al., 2023; Montagna et al., 2023; Reizinger et al., 2024a). Thus, the real question is whether they are *realistic* in practical RL scenarios? Based on empirical evidence from CSF, our answer is affirmative. As in practice, one might not have access to the true states, we formulate these statements in terms of the learned features $\phi(o)$.

(i) In CSF skills are drawn uniformly from the hypersphere, and if we have sufficiently many of them, then they are "diverse" enough to almost surely form an affine generator system $\mathbb{R}^d$. We demonstrate that this setup is sufficient, but not necessary: a set of discrete skills also leads to high identifiability scores (Fig. 4). Also, empirical observations show that the skill-conditioned policy cannot be optimal if it does not depend on the skill, i.e., if it has maximum entropy.
(ii) Empirical observations show that the features of consecutive observations $\phi(o), \phi(o')$ are close to each other on the hypersphere (Zheng et al., 2025, Fig. 2(a-b)).[1]
(iii) Empirical observations show the learned feature differences $\phi(o') - \phi(o)$ are uniformly distributed on the hypersphere (Zheng et al., 2025, Fig. 2(c))
(iv) In practice, MISL methods are either trained from directly observing the state or from pixels. Both cases reasonably fulfill the assumption.
(v) The neural networks used for training use an inner product parametrization, and empirical evidence by Zheng et al. (2025) showed that this works the best in practice.

**Identifiability of the underlying features.** Under Assum. 1, the feature differences are identified up to a linear map (we defer the formal statement to Appx. B.2):

**Proposition 1** (Identifiability of CSF feature differences (informal)). *Under Assum. 1, when a continuous encoder and a linear classifier $Z$ globally minimize the cross-entropy objective* (1)*, then the state differences $s' - s$ are identified up to a linear map $\mathbf{A} \in \mathbb{R}^{d \times d}$, i.e., $\phi(o') - \phi(o) = \mathbf{A}[s' - s]$.*

---

[1]Zheng et al. (2025) evaluate the representations learned with METRA. We also verify with CSF in all environments we evaluate on in Appx. D.4.

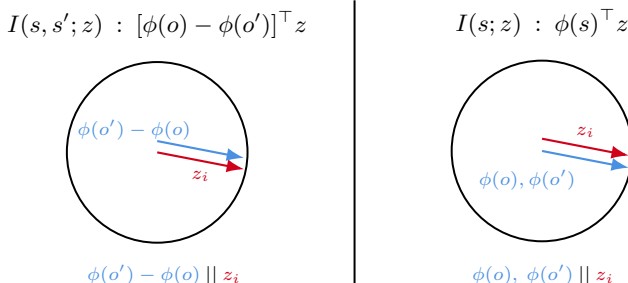

Figure 2: **Illustrating the advantage of parametrizing mutual information as $I(s, s'; z_i)$ vs.** $I(s; z_i)$: the reward function and the feature learning objective impose different inductive biases on the structure of the latent space. **Left:** using feature $\phi(o') - \phi(o)$ to parametrize $I(s, s'; z_i)$ ensures that the embeddings of consecutive states are close but distinct. **Right:** optimizing $I(s; z)$ incentivizes the embeddings of consecutive states to both be parallel to skill $z_i$, which can lead to collapsed features (i.e., $\phi(o) = \phi(o')$).

The proof follows by the straightforward application of the proof technique of Reizinger et al. (2024a, Thm. 1)—in Appx. B.2, we provide a high-level overview thereof. Since the linear map in Prop. 1 is the same for all (unit-normalized) state differences and the model has an inner-product parametrization, linear identifiability also holds for the states. We formalize this statement in Prop. 3 and defer it to Appx. B.2.

These results mean that the features learned by CSF (Zheng et al., 2025) will correspond to the ground-truth states of the underlying POMDP up to a linear transformation. Having identifiability both for $s$ and $(s' - s)$ might suggest that it does not matter whether the objective optimizes a lower bound on $I(s, s'; z)$ or $I(s; z)$. As we show in § 4.2, the difference lies in the additional geometric constraints on the latent space. Namely, there exist spurious solutions of the InfoNCE objective that do not preserve the structure of the latent space (Wang et al., 2022).

## 4.2 Insights from ICA theory

**Mutual information formulation matters for the geometry of feature space.** In the literature, there are many choices for optimizing MI, namely, $I(s, s'; z)$ versus $I(s_0, s; z)$ versus $I(s; z)$—cf. Fig. 2 for a comparison. By looking into the policy and analyzing what maximizing $\mathbb{E}_{s,z,a} [\phi(o') - \phi(o)]^\top z$ means, we hope to shed light on the advantages of $I(s, s'; z)$ over the latter versions. Maximizing the inner product $[\phi(o') - \phi(o)]^\top z$ means that the difference $\phi(o') - \phi(o)$ needs to be parallel to $z$. This implies that neither $\phi(o)$ nor $\phi(o')$ can collapse to the same vector, as they need to be distinct such that their difference is parallel to $z$. On the other hand, optimizing $I(s; z)$ would mean two separate conditions for $\phi(o)$ and $\phi(o')$. But if both are parallel to $z$, then they must be parallel, i.e., the representation collapses. This violates the assumption that consecutive states should be close to each other in embedding space (but neither the same, nor very far apart, cf. Assum. 2(ii)). The same argument holds for $I(s_0, s; z)$ with the difference of offsetting the whole space by the initial state. Instead, $I(s, s'; z)$ prescribes the "closeness" of consecutive states.

**A practical implication of diversity rewards.** A perhaps interesting interpretation of rewards such as (3) is that it quantifies a notion of data diversity—for other means to measure diversity, refer to (Reizinger et al., 2024b). From a theoretical perspective, diversity is a binary question, as it is required to make a matrix invertible. But this matrix can also be ill-conditioned without being rank-deficient, leading to performance deterioration (Rajendran et al., 2023). Understanding when the reward is a good predictor of learning useful representations from a given data set (e.g., in offline RL) is an interesting avenue for future work.

**Maximum entropy policies lead to worse performance.** Although several RL methods use an entropy regularizer (Eysenbach et al., 2019; Sharma et al., 2020; Park et al., 2024b), a too strong regularization can lead to worse performance. Entropy regularization, in this extreme, is also not suitable for skill diversity, as a maximum entropy policy breaks the dependence on the skill of the skill-conditioned policy. We state this observation from the literature formally in Appx. B.3. Intuitively, if the actions—and, thus, the state transitions—do not depend on the skill, then from a given state pair $(s, s')$ it is impossible to infer the skill with the discriminative model $q(z|s, s')$, and the reward cannot be optimal.

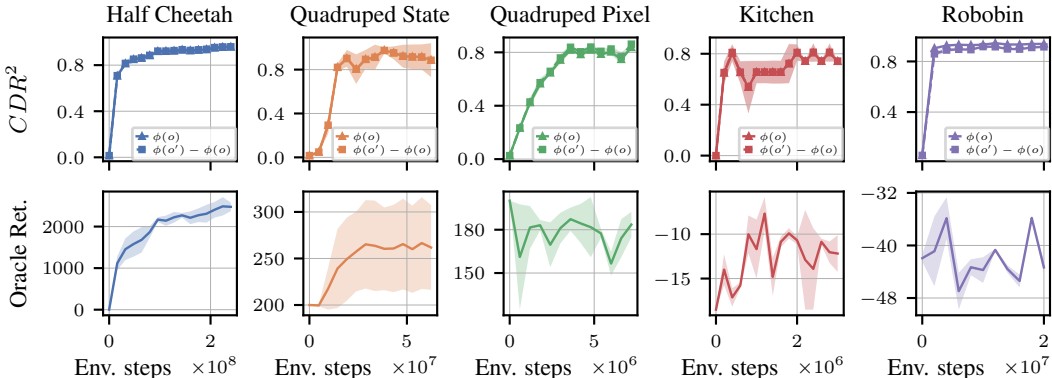

Figure 3: **CSF explores the state space and identifies the underlying states in MuJoCo and DMC, up to a linear transformation. Top:** coverage-dependent $R^2$ score, measuring both state exploration and linear identifiability for both features $\phi(o)$ and feature differences $\phi(o') - \phi(o)$ (higher is better); **Bottom:** oracle return indicating zero-shot task transfer. Error bars represent one standard deviation.

## 5 EXPERIMENTS

**Setup.** We use the codebase of Zheng et al. (2025) and run experiments in the MuJoCo and Deep-Mind Control (DMC) environments with the CSF algorithm. During self-supervised pretraining, we monitor the learned (successor) features. To evaluate identifiability, we set the feature dimensionality to match the number of ground-truth features. We use either states (Half Cheetah, Ant, Quadruped State) or pixels (Quadruped Pixel, Kitchen, Robobin) as observations to learn the features $\phi(o)$. In the state-based environments, the encoder is, in principle, able to represent the identity map; however, it is unclear whether such features would optimize the loss. Unless otherwise noted, error bars represent one standard deviation. Refer to Appx. C for details.

**Metrics.** The benefit of using simulated MuJoCo and DMC environments is having access to the ground truth states, enabling one to evaluate the relationship between the learned features $\phi(o)$ and the ground-truth latent factors $s$. To measure exploration, we report the **state coverage,** i.e., the unique states observed across all evaluation trajectories using different skills. To assess downstream performance, we also report the **oracle return** from the best rollout among the sampled skills, i.e., the extrinsic reward defined by the environment. The oracle return evaluates *zero-shot* skill transfer: a diverse skill set should naturally contain one that performs well on the downstream task. As the identifiability guarantees (Prop. 1) hold up to a linear transformation, we fit a linear map $\mathbf{A}$ between the features $\phi(o)$ and the ground truth states $s$ by minimizing $\|s - \mathbf{A}\phi(o)\|_2^2$ and report the coefficient of determination $R^2$ (Wright, 1921) of the linear fit, which is the standard metric in the ICA literature (Hyvarinen & Morioka, 2016; Hyvarinen et al., 2019; 2023; Hyvärinen et al., 2023). However, the $R^2$ score can be misleading in RL: it measures performance only on the *visited* states and not *all* states; thus, it cannot distinguish between a collapsed scenario (i.e., no exploration) and a well-explored one. Thus, we introduce a relative coverage-dependent $R^2$ score ($CDR^2$), defined as the harmonic mean of the normalized coverage and the $R^2$ score. This yields a metric which is near zero if an agent has *either* a low $R^2$ score *or* poor coverage. For details, refer to Appx. C.3.

**Results.** We investigate whether CSF identifies the ground-truth latent states up to a linear transformation and how this relates to the oracle return. We report scores in Fig. 3 across two state-based MDPs (Cols. 1-2; the Ant results are in Appx. D) and three pixel-based environments (Cols. 3-5). CSF both explores the state space and identifies the ground-truth states up to a linear transformation (Fig. 3, top; separate $R^2$ and coverage results are in Figs. D.1 and D.2), aligning with Prop. 1. In the state-based environments, it is by no means obvious that the encoder will keep all information about the underlying states, e.g., there could have existed a shortcut solution that optimizes the CSF objective while discarding some information about the states. In state-based environments, the oracle return and identifiability performance ($CDR^2$) are strongly correlated. In pixel-based environments, the relationship is less clear, though the agents both explore and learn to extract the ground-truth states, even though the oracle return is noisier. We provide a correlation analysis in Tab. D.1.

To test the role of the diversity conditions in nonlinear ICA (Hyvarinen et al., 2019; Rajendran et al., 2023; Wendong et al., 2023; Reizinger et al., 2024b), we investigate how skill diversity affects state

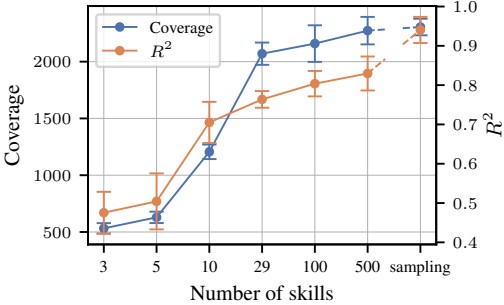

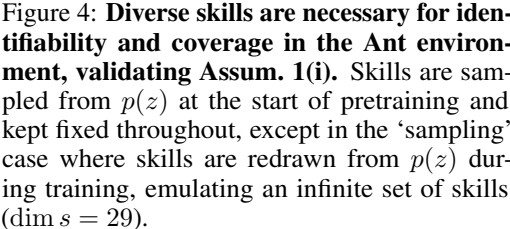

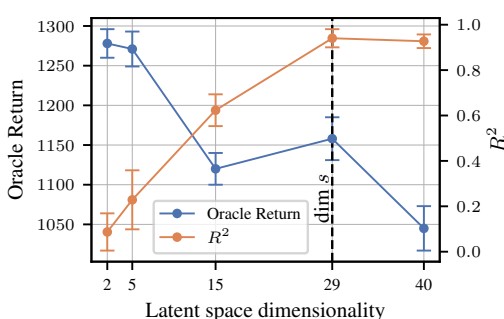

Figure 4: **Diverse skills are necessary for identifiability and coverage in the Ant environment, validating Assum. 1(i).** Skills are sampled from $p(z)$ at the start of pretraining and kept fixed throughout, except in the 'sampling' case where skills are redrawn from $p(z)$ during training, emulating an infinite set of skills ($\dim s = 29$).

Figure 5: **Sufficient latent space dimensionality is necessary for identifiability, validating Assum. 1(v).** Linear identifiability requires that the feature space has at least as many dimensions as the true state. However, a smaller latent space is more beneficial for task transfer.

identifiability and coverage. Namely, Prop. 1 only requires that skills span $\mathbb{R}^d$, suggesting that a limited set of skills might be sufficient to achieve identifiability. We vary the number of skills $z_i$ and use a fixed set of $z_i$ vectors. This scheme is compared to the original version of sampling each time from the uniform $p(z)$. Using a small fixed set of pre-sampled skills is insufficient to cover the state space during pretraining or to yield identifiable representations, as shown by the peak coverage and corresponding $R^2$ score in Fig. 4. Appx. C shows further results for zero-shot task transfer. We also investigate how latent space dimensionality affects both identifiability and zero-shot task transfer—if the feature space is lower dimensional than the state space in the POMDP, then all state information might be present but not linearly decodable. Fig. 5 shows $R^2$ scores and zero-shot task transfer performance for varying latent space dimensionality. A low-dimensional latent space yields an information bottleneck that prevents the feature differences from encoding the ground truth states linearly; however, CSF still performs well on task transfer[2]. Notably, *__this result does not imply that CSF is not able to extract the ground-truth states, it only means that it is not able to do it linearly__*—indeed, works studied representations with different geometries (Csordás et al., 2024); however, both theoretical understanding and corresponding metrics are missing. We defer how latent dimensionality affects state coverage to Appx. C.

# 6    DISCUSSION

**Limitations.**    We connected identifiability insights from nonlinear Independent Component Analysis (ICA) to mutual information skill learning (MISL) and formulated practical insights. Our result relies on the observations of Zheng et al. (2025) and holds across multiple state- and pixel-based environments (for verification, refer to Appx. D.4). However, it requires further research to determine whether the technical assumptions can be relaxed to incorporate a broader range of environments.

**Extension to related works.**    Our work is of an explanatory nature, advancing our understanding of mutual information skill learning (MISL) methods in Reinforcement Learning (RL). To the best of our knowledge, we are the first to prove identifiability of the learned features in RL (Prop. 1), particularly, for Contrastive Successor Features (CSF) (Zheng et al., 2025). Nonetheless, as the key components for state identifiability are shared across many MISL methods, we expect our results to hold more generally. This insight helped explain why particular design choices in MISL are successful, including the inner product parametrization of the critic and also shed light on how different mutual information formulations affect the learned representation (§ 4.2).

**Conclusion.**    Our work theoretically proves that learning diverse skills is a meaningful surrogate objective in Reinforcement Learning (RL) for learning the ground-truth states of the environment up to a linear transformation. We show this by connecting the mutual information skill learning

---

[2]CSF is designed for lower feature space dimensionality, and we empirically found it to be sensitive to increasing this hyperparameter

(MISL) family to nonlinear Independent Component Analysis (ICA) methods, and proving linear identifiability for the features learned by Contrastive Successor Features (CSF) (Zheng et al., 2025). Our identifiability guarantees not only provide a possible explanation of why MISL works, but also identify the key components of successful MISL methods. Furthermore, our theoretical insights help elucidate some failure modes of previous methods. We validated our theoretical claims empirically in DMC and MuJoCo environments, showing that CSF simultaneously explores the state space and learns features that identify the ground-truth states up to a linear transformation. As exploration and state identification showed a positive correlation with the extrinsic oracle return, defined by each environment, this suggests that identifiability can be helpful for zero-shot task transfer. However, as it is well known in the identifiability literature, in some cases recovering more latent factors (i.e., better identifiability score) can be at odds with downstream performance on a particular task (Rusak et al., 2025). We hope that our insights will open up new research possibilities and also help practical algorithm design.

REPRODUCIBILITY

We provide shell scripts to reproduce our results in the `scripts` folder of our GitHub repository. To obtain error bars, we additionally varied the seed parameter in the scripts to obtain three to five independent runs.

ACKNOWLEDGEMENTS

The authors thank Jens Tuyls for his help regarding the MISL experiments. Patrik Reizinger and Bálint Mucsányi acknowledge their membership in the European Laboratory for Learning and Intelligent Systems (ELLIS) PhD program and thank the International Max Planck Research School for Intelligent Systems (IMPRS-IS) for its support. This work was supported by the German Federal Ministry of Education and Research (BMBF): Tübingen AI Center, FKZ: 01IS18039A. Wieland Brendel acknowledges financial support via an Emmy Noether Grant funded by the German Research Foundation (DFG) under grant no. BR 6382/1-1 and via the Open Philanthropy Foundation funded by the Good Ventures Foundation. Wieland Brendel is a member of the Machine Learning Cluster of Excellence, EXC number 2064/1 – Project number 390727645. This research utilized compute resources at the Tübingen Machine Learning Cloud, DFG FKZ INST 37/1057-1 FUGG.

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

## A   IMPACT STATEMENT

This paper presents work whose goal is to advance the fields of Reinforcement Learning and Identifiable Representation Learning. Our focus on representation identifiability promotes transparency and interpretability, which are important safeguards against unintended use.

## B   PROOFS

### B.1   FORMAL ASSUMPTIONS AND FEASIBILITY

**Definition 2** (Affine Generator System (Reizinger et al., 2024a) (Defn. 1) ). *A system of vectors $\left\{z_i \in \mathbb{R}^d\right\}$ is called an* affine generator system *if any vector in $\mathbb{R}^d$ is an affine linear combination of the vectors in the system. Put into symbols: for any $z_i \in \mathbb{R}^d$ there exist coefficients $\alpha_i \in \mathbb{R}$, such that*

$$z = \sum_i \alpha_i z_i \quad and \quad \sum_i \alpha_i = 1. \tag{4}$$

**Lemma 1** (Properties of affine generator systems (Reizinger et al., 2024a) (Lem. 1)). *The following hold for any affine generator system $\left\{z_i \in \mathbb{R}^d\right\}$:*
*1. for any $i \neq j$ the system $\{z_i - z_j\}$ is now a generator system of $\mathbb{R}^d$;*
*2. the invertible linear image of an affine generator system is also an affine generator system.*

**Assumption 2** (Adapted from (Reizinger et al., 2024a) (Assm. 1C)). *For consecutive states $s, s' \in \mathbb{R}^d$, $(s' - s) \in \mathcal{S}^{d-1}$ and skills $z_i \in \mathcal{S}^{d-1}$, we assume:*
  *(i) The finite set of skills is unit-normalized and forms an affine generator system of $\mathbb{R}^d$ (Defn. 2). Each pair of features corresponds to one skill. That is, an ideal discriminator can uniquely map $(s' - s) \mapsto z$, yielding diverse skills (cf. Defn. 1).[3]*
  *(ii) The skill-conditioned features $p(s' - s|z)$ follow a von Mises-Fisher (vMF) distribution on the hypersphere with mean $z$, expressing that consecutive states are close to each other:*

$$(s' - s) \sim p(s' - s|z) \propto e^{\kappa \langle z, s'-s \rangle}. \tag{5}$$

  *(iii) The marginal of the features $(s' - s)$ is uniform on the hypersphere.*
  *(iv) The critic $q(z|\phi(o), \phi(o'))$ uses an encoder $\phi : \mathbb{R}^D \to \mathbb{R}^d$ to learn the features with an inner product parametrization $[\phi(o') - \phi(o)]^\top z_i$, it optimizes a contrastive objective (1), and is expressive enough to reach the global optimum of the objective.*
  *(v) The observations $o$ are generated by passing the latent state $s$ through a continuous and injective generator function $g \colon \mathcal{S}^{d-1} \to \mathbb{R}^D$, i.e., $o = g(s)$, where $D \geq d$.*

*Assumption feasibility.* As we are modeling a practical scenario, we need to investigate whether these assumptions are realistic:
  (i) In CSF skills are drawn uniformly from the hypersphere, and if we have sufficiently many of them, they span $\mathbb{R}^d$ almost surely, thus they almost surely form an affine generator system. We demonstrate that this setup is sufficient, but not necessary: a set of discrete skills also leads to high identifiability scores (Fig. 4). The policy optimizes $[\phi(o') - \phi(o)]^\top z_i$. As we show in Prop. 4, a maximum entropy policy is not diverse. Applying our argument to pairs of skills, one can always improve diversity if the state transitions depend on any one of those two skills (as opposed to no skill dependence, which is implied by a maximum entropy policy), which implies that each state transition can be mapped to a single skill. Refer to Prop. 4 for details.
  (ii) Empirical observations show that the learned features $\phi(o') - \phi(o)$ follow a vMF for CSF (Zheng et al., 2025, Fig. 2(a-b)). Our empirical validation also verifies this assumption for Half Cheetah, Kitchen, Quadruped, and Robobin in Appx. D.4.
  (iii) Empirical observations show the learned features $\phi(o') - \phi(o)$ follow a uniform marginal on the hypersphere for CSF (Zheng et al., 2025, Fig. 2(c))
  (iv) The neural networks used for training use an inner product parametrization, and empirical evidence by (Zheng et al., 2025) showed that this works the best in practice.
  (v) In practice, MISL methods are either trained from directly observing the state (i.e., $g$ is the identity) or from pixels. Both cases reasonably fulfill the assumption.  □

---

[3]Different pairs of consecutive states can map to the same skill

## B.2 IDENTIFIABILITY OF FEATURE DIFFERENCES

**Proposition 2** (Identifiability of CSF feature differences)**.** *When Assum. 2 holds and a continuous encoder $\phi\colon \mathbb{R}^D \to \mathbb{R}^d$ and a linear classifier $Z$ globally minimize the cross-entropy objective* (1)*, then the composition $h = \phi \circ g$ is a linear map from $\mathbb{S}^{d-1}$ to $\mathbb{R}^d$, i.e., $\phi(o') - \phi(o) = \mathbf{A}\,[s' - s]$ where $\mathbf{A} \in \mathbb{R}^{d \times d}$ is a linear map.*

*Proof.* The proof follows by the straightforward application of the proof technique (Reizinger et al., 2024a, Thm. 1). $\qquad\square$

To provide a high-level overview of how we have arrived at this result, we state our thought process:
1. We establish that the assumptions from Reizinger et al. (2024a) hold for CSF;
2. Our insight to connect the contrastive objective to the state differences is that the CSF loss has the same parametrization as the one in (Reizinger et al. (2024a) ) with the substitution $z = (s - s')$ ($z$ is the notation of Reizinger et al. (2024a), and ***does not refer to skills***);
3. Then we apply the identifiability result of (Reizinger et al. (2024a) ), which proceeds by arguing about the Bayes optimum of cross entropy minimization. For this, the important substeps are:
    (a) Exploiting the symmetries of the inner product parametrization;
    (b) Using skill diversity (required to construct an invertible matrix to solve a linear equation system to express the learned features in terms of the ground-truth states;
4. This gives us linear identifiability of $z = (s - s')$ , yielding the formula $\mathbf{A}(s - s')$

The linear map in Prop. 2 is the same for all (unit-normalized) states. Combined with the linear parametrization, this implies that the feature differences are also identified up to a linear transformation:

**Proposition 3** (Feature identifiability in CSF)**.** *Prop. 2 implies by the inner product parametrization of $q(z|s, s')$ that $\phi(o) = \mathbf{A}s$ and $\phi(o') = \mathbf{A}s'$ with the same $\mathbf{A}$, thus, the features are also identified up to a linear transformation.*

*Proof.* The proof follows from the linear parametrization of the model. By linear identifiability of the feature differences (as they lie on $\mathcal{S}^{d-1}$, an offset is not possible), we have
$$\phi(o') - \phi(o) = \mathbf{A}\,[s' - s] = [\mathbf{A}s' - \mathbf{A}s]$$

$\qquad\square$

## B.3 SUBOPTIMALITY OF THE MAXIMUM ENTROPY POLICY

**Proposition 4.** *[A maximum-entropy policy in CSF is not diverse] A maximum entropy skill-conditioned policy $\pi(a|o, z) = $ Uniform is not diverse and cannot maximize the reward $\mathbb{E}_{s,z,a}\,[\phi(o') - \phi(o)]^{\top} z$.*

*Indirect.* Fix the initial state and assume that the skill-conditioned policy has maximum entropy, i.e., it follows a uniform distribution and maximizes the reward—given that the policy network is sufficiently flexible to express such a policy. This implies that the expectation over the actions does not depend on $z$, yielding in expectation the same $[\phi(o') - \phi(o)]$ for each skill.

$$r_z(\phi(o), \phi(o')) = \mathbb{E}_{s,z,a}\,[\phi(o') - \phi(o)]^{\top} z \tag{6}$$

$$= \int_{s,z,a} [\phi(o') - \phi(o)]^{\top} z\, p(s'|s, a)\pi(a|o, z)p(z)p(s)\,ds\,dz\,da \tag{7}$$

In this case, the skill-conditioned policy becomes independent of $z_i$, as the uniform distribution over the action space has maximum entropy. Substituting $\pi(a|o, z) = \pi(a|o)$ yields

$$= \int_{s,z,a} [\phi(o') - \phi(o)]^{\top} z\, p(s'|s, a)\pi(a|o)p(z)p(s)\,ds\,dz\,da \tag{8}$$

and by reordering the terms, we get

$$= \int_{s,z,a} [\phi(o') - \phi(o)]^{\top} z\, \pi(a|o)p(s'|s, a)p(s)\,ds\,da\,p(z)\,dz. \tag{9}$$

Note that $p(s'|s, a)$, $\pi(a|o)$, and $p(s)$ are independent of $z_i$, thus, $[\phi(o') - \phi(o)]$ are also independent of $z_i$. As a skill vector parallel to $[\phi(o') - \phi(o)]$ maximizes the inner product, and the skills are on the unit hypersphere, this yields a unique solution. However, as the skills are drawn uniformly from the unit hypersphere, they are distinct. That is, the inner products will differ for $z_i \neq z_{j \neq i}$. Thus, both skills cannot maximize the reward, leading to a contradiction. $\qquad\square$

## C    EXPERIMENTAL DETAILS

### C.1    COMPUTE RESOURCES

All experiments were run in a compute cluster using an Intel Xeon Gold CPU (16 cores, 2.9 GHz) and NVIDIA RTX 2080 Ti GPUs and used at most $48$ GB of RAM. No experiment required more than 3 days of runtime. In total, our experiments took $0.4$ GPU years.

### C.2    HYPERPARAMETER SEARCH

To train the CSF method in the Ant, Half Cheetah, Quadruped State, and Quadruped Pixel environments, we used the hyperparameters included in the GitHub repository of Zheng et al. (2025) as a starting point and modified (i) the encoder $\phi$'s backbone architecture to aid identifiability by introducing skip-connections, (ii) the latent space dimensionality to match the ground truth state's dimensionality, (iii) the trade-off factor $\xi$ between the two factors of the contrastive loss (Zheng et al., 2025, Eq. (10) & paragraph below), and (iv) the number of negative samples in the contrastive loss. We found the CSF method to be sensitive to the latent space dimensionality, but an appropriate choice of $\xi$ mitigated this sensitivity and encouraged learning. Increasing the number of negative samples in the contrastive loss led to performance gains in some environments (w.r.t. state space coverage and oracle return), which is intuitively explained by requiring more samples to cover the latent space well. In our experiments that varied the number of skills in the discrete skill set, we kept all other hyperparameters fixed to the continuous sampling case. In the experiments varying the latent space dimensionality, the other hyperparameters were fixed to those of dim $= 29$.
For the exact hyperparameter configurations, see the code in the supplementary material.

### C.3    EVALUATION METRICS DETAILS

**Coverage.**    We define our coverage indicators following Zheng et al. (2025). In the Ant, Half Cheetah, Quadruped, and Robobin environments, we define coverage as the number of unique integer-discretized coordinate vectors observed across all evaluation trajectories (using different skills). In the Kitchen environment, we define coverage as the sum of binary success indicators across all six kitchen tasks (BottomBurner, LightSwitch, SlideCabinet, HingeCabinet, Microwave, Kettle). Intuitively, the agent requires a diverse skillset to achieve broad coverage.

**Relative coverage.**    To make coverage comparable across environments, we define relative coverage as the coverage normalized by (an estimate of) the best attainable coverage. We use an empirical estimate by choosing the maximum coverage from either our experiments or from coverage results in the literature (when available).

**Oracle return.**    We define the oracle return as the highest cumulative discounted return attained across all evaluation trajectories. Intuitively, this is the return the skill-conditioned agent would achieve if an oracle shared the best skill for the task before the rollout. In a meaningfully diverse skillset, it is likely that there exists a skill that solves the task defined via the (PO)MDP.

**Coverage-dependent $R^2$ score ($CDR^2$).**    A limitation of the $R^2$ score is that it can obtain a high value even if the agent has not explored the environment well. Namely, if the ground-truth latents from all visited states are recovered up to a linear transformation *but* the visited states only form a small subset of *all* states, then we cannot expect good downstream performance from our agent. To make the $R^2$ score distinguish between collapsed (no exploration) and well-explored scenarios, we introduce a coverage-dependent $R^2$ score, which is defined as the harmonic mean of the relative coverage ($\tilde{C} := \frac{C}{C_{\max}}$ where $C$ is the coverage score and $C_{\max}$ is the highest attainable coverage in the environment) and the $R^2$ score. That is,

$$CDR^2 = \frac{2\tilde{C}R^2}{\tilde{C} + R^2}.$$

We choose the harmonic mean and not the arithmetic or geometric means, as we want to assign a low score to an agent that has *either* a low $R^2$ score *or* poor coverage, and out of the three means, the harmonic mean punishes low values most severely. If either component is near zero, the harmonic mean will be near zero regardless of how high the other component is.
Our plots showing the $CDR^2$ score also serve as a comparison against a *random baseline*, demonstrating the shortcoming of $R^2$. Namely, at the start of training, $R^2$ might be high—as the agent has not explored the state space—but $CDR^2$ is close to zero.

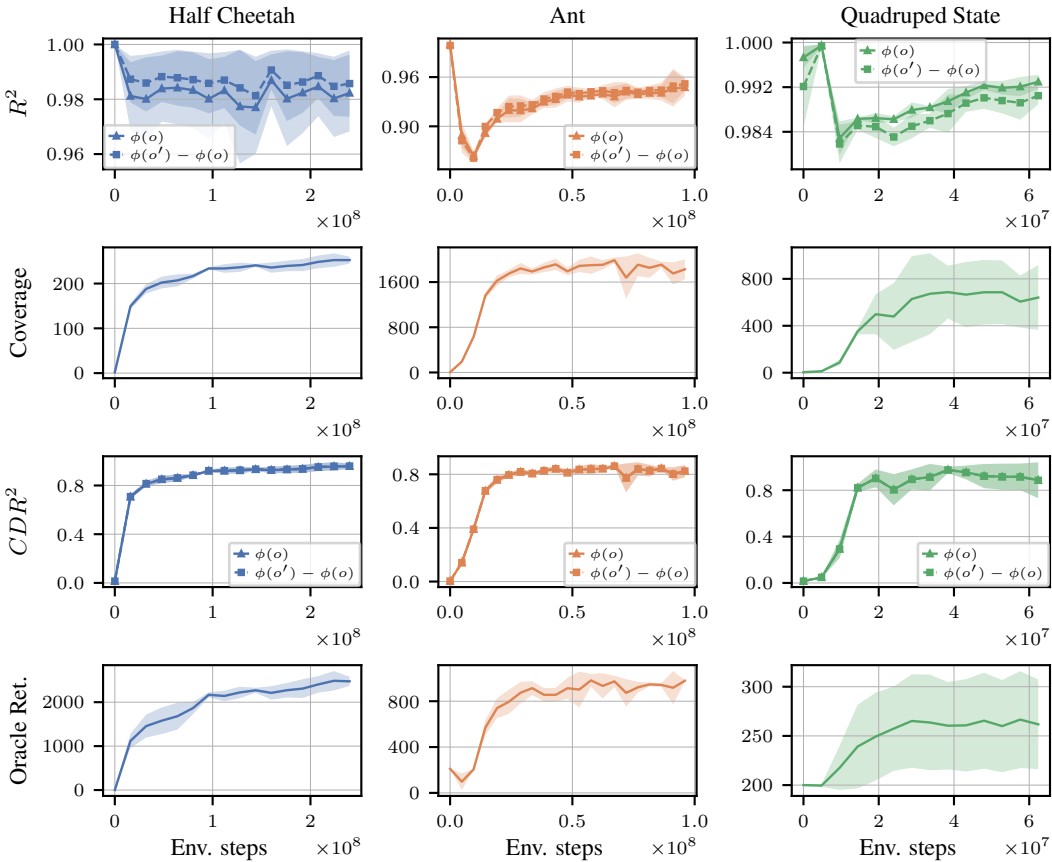

Figure D.1: **CSF identifies the underlying states in MuJoCo and DMC up to a linear transformation. First:** Identifiability of both features $\phi(o)$ and feature differences $\phi(o') - \phi(o)$, measured by the $R^2$ score (higher is better); **Second:** state coverage, indicating exploratory behavior; **Third:** coverage-dependent $R^2$ score, measuring both state exploration and state identifiability; **Fourth:** oracle return indicating zero-shot task transfer performance. Error bars represent one standard deviation.

### C.3.1 NUMERICAL INSTABILITY IN CALCULATING $R^2$ SCORES

We investigated the discrepancy between the raw $R^2$ score of embeddings $\phi(o)$ and single-step embedding differences $[\phi(o') - \phi(o)]$, and found that the reason is *numerical instability*, which is caused by two factors:

1. some dimensions in the ground-truth states having no or extremely low variance during a rollout, leading to an extreme sensitivity in the $R^2$ score even in double precision, and
2. the neighboring states having extremely small differences in some environments, leading to small numerical imprecisions inflicting significant relative errors.

We have resolved (i) by filtering constant or extremely low-variance ($1 \cdot 10^{-8}$) state dimensions, and (ii) by calculating the differences over multiple time steps in some environments for stable and meaningful results. Fig. 3 applies these fixes to the $R^2$ scores.

## D FURTHER RESULTS

### D.1 CORRELATION ANALYSIS FOR PERFORMANCE METRICS

Tab. D.1 shows the shortcoming of $R^2$ distinguishing collapsed and well-explored scenarios *(R² vs Coverage)*; furthermore, it is also not indicative of the oracle return *(R² vs Oracle)*. However, $CDR^2$ is more indicative of the oracle return, especially in pixel-based environments *(CDR² vs Oracle)*.

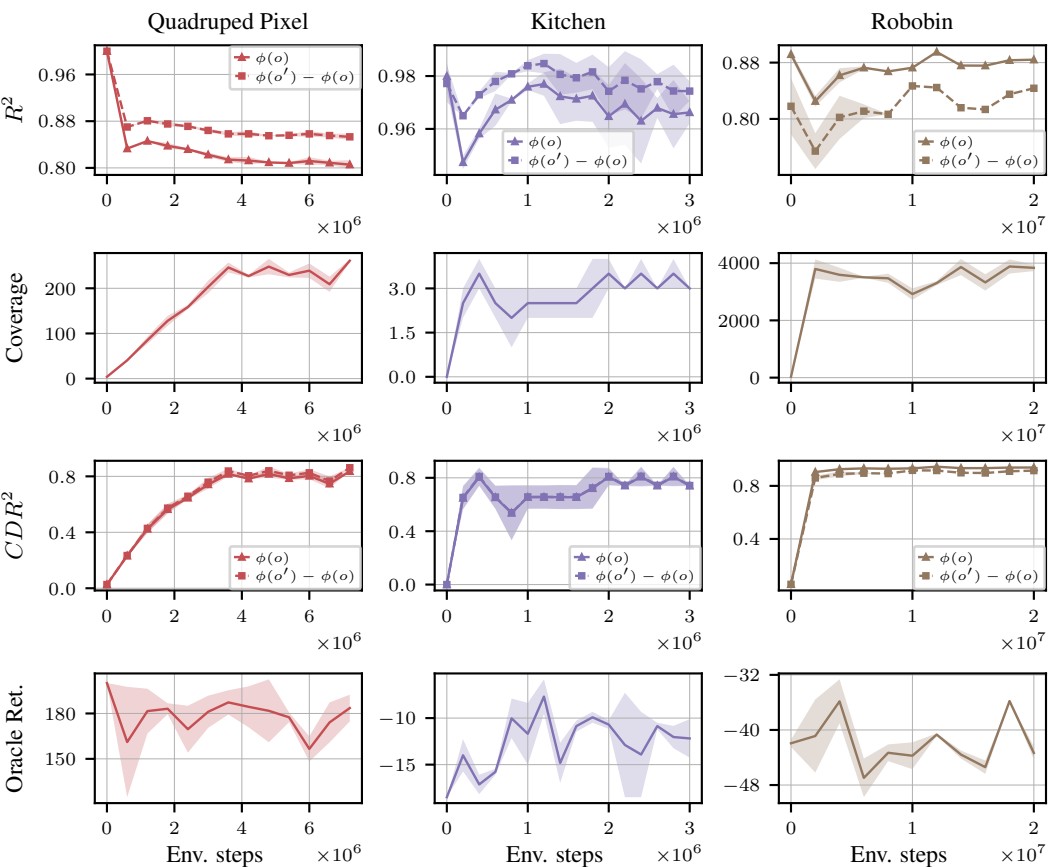

Figure D.2: **CSF identifies the underlying states in MuJoCo and DMC up to a linear transformation. First:** Identifiability of both features $\phi(o)$ and feature differences $\phi(o') - \phi(o)$, measured by the $R^2$ score (higher is better); **Second:** state coverage, indicating exploratory behavior; **Third:** coverage-dependent $R^2$ score, measuring both state exploration and state identifiability; **Fourth:** oracle return indicating zero-shot task transfer performance. Error bars represent one standard deviation.

| Environment | $CDR^2$ vs Oracle | $R^2$ vs Oracle | $R^2$ vs Coverage |
|---|---|---|---|
| Half Cheetah | 0.9261 | $-0.7095$ | $-0.7868$ |
| Ant | 0.9441 | 0.3465 | 0.1331 |
| Quadruped State | 0.9673 | $-0.3623$ | $-0.2395$ |
| Quadruped Pixel | $-0.2068$ | 0.5348 | $-0.7623$ |
| Kitchen | 0.4125 | 0.2092 | $-0.5005$ |
| Robobin | $-0.0046$ | $-0.0708$ | $-0.3636$ |

Table D.1: **Correlation analysis of the pairs of metrics from Figs. D.1 and D.2**: the results pinpoint the insufficiency of $R^2$ to predict good zero-shot performance on the oracle return, as it does not consider the coverage of the state space. $CDR^2$, on the other hand, is more indicative of the oracle return, especially in state-based environments.

## D.2 THE EFFECT OF SKILL DIVERSITY ON ZERO-SHOT PERFORMANCE

Fig. D.3 shows the counterpart of Fig. 4, where the zero-shot skill transfer performance is evaluated instead of state space coverage. The result indicates that Assum. 2(i), i.e., having a sufficiently diverse set of skills, seems to be necessary for good downstream performance.

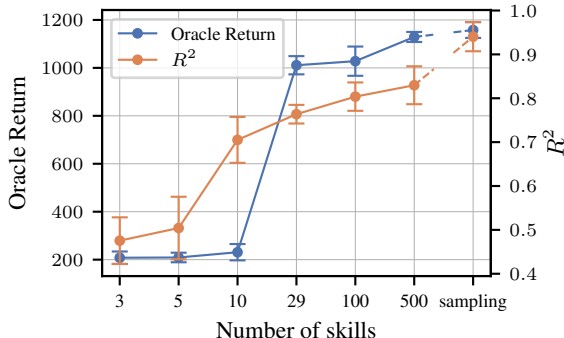

Figure D.3: **The effect of skill diversity on state identifiability and zero-shot skill transfer in the Ant environment.** Skills are sampled from $p(z)$ at the start of pretraining and kept fixed throughout, except in the 'sampling' case where skills are redrawn from $p(z)$ during training, emulating an infinite set of skills. An insufficient number of skills violates Assum. 2(i), leading to both weaker zero-shot skill transfer and a lower $R^2$ score.

## D.3 THE EFFECT OF LATENT SPACE DIMENSIONALITY ON COVERAGE

Fig. D.4 shows a variant of Fig. 5 with state space coverage reported instead of zero-shot skill transfer performance. For linear identifiability, it is required that the feature space has at least as many dimensions as the ground-truth states—as required by Assum. 2(v). This does not necessarily mean that the features do not capture (all) information about the ground-truth states; it only means that the information cannot be decoded linearly to reconstruct all ground-truth states. With increasing latent dimensionality, coverage tends to decrease, primarily because the CSF algorithm was designed for lower latent dimensionality.

## D.4 MARGINAL G-TESTS AND HISTOGRAMS FOR VERIFYING FEATURE CONCENTRATION ON THE HYPERSPHERE

We have verified Assum. 2(ii) for all environments we evaluate on using a series of G-tests on the marginal histograms (i.e., on the 1D histograms corresponding to individual dimensions in the latent space). We tested for Gaussianity, as high-dimensional Gaussian random variables concentrate on the corresponding hypersphere. We then investigated whether the number of nominal 5% rejections was larger than chance would produce if all marginals were truly Gaussian (via a one-sided binomial test with rate 0.05). In all environments except Half Cheetah, this excess-rejection test was not significant (all $p >= 0.05$), meaning we found no environment with a violation of Gaussianity. In Half Cheetah, we observed some latent dimensions with bimodal marginals next to Gaussian-like ones.

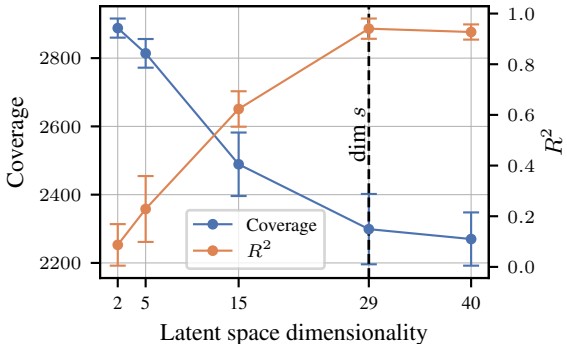

Figure D.4: **The effect of latent space dimensionality on state identifiability and coverage in the Ant environment.** Linear identifiability requires that the feature space has at least as many dimensions as the true state (cf. Assum. 2(v)).

We include 4-4 histograms of randomly sampled feature dimensions for the Half Cheetah (Fig. D.7), Kitchen (Fig. D.5), Quadruped (Fig. D.6), and Robobin (Fig. D.8) environments.[4] These figures, excluding the single exception of feature dimension 3 in the Half Cheetah environment (Fig. D.7), indicate that Assum. 2(ii) holds across multiple environments.

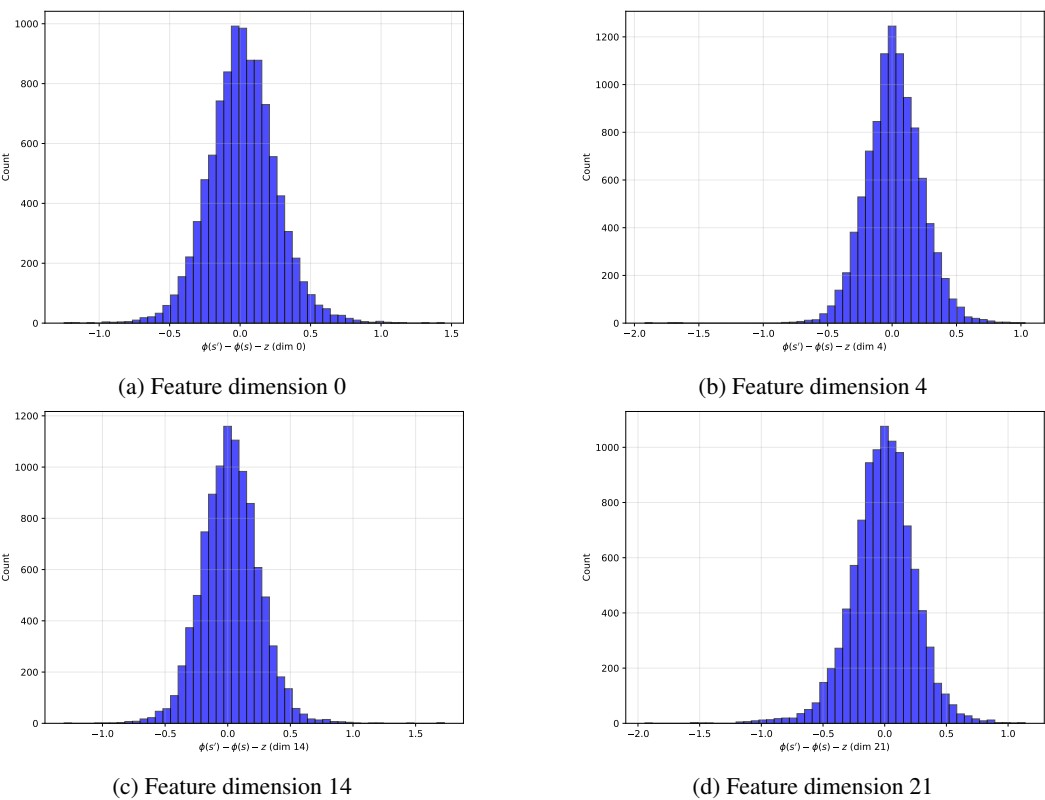

(a) Feature dimension 0

(b) Feature dimension 4

(c) Feature dimension 14

(d) Feature dimension 21

Figure D.5: **Marginal histograms for four randomly sampled feature dimensions in the *Kitchen* environment:** the histograms clearly show that Assum. 2(ii) holds across multiple environments.

## D.5 IDENTIFIABILITY OF OBJECTS WITHIN STATES

To measure whether the representation captures the task-relevant objects (instead of only the agent's proprioception, which is easier to reconstruct), we also report an "object $R^2$ score", computed on the

---

[4]Refer to (Zheng et al., 2025, Fig. 2) for histograms in the Ant environment.

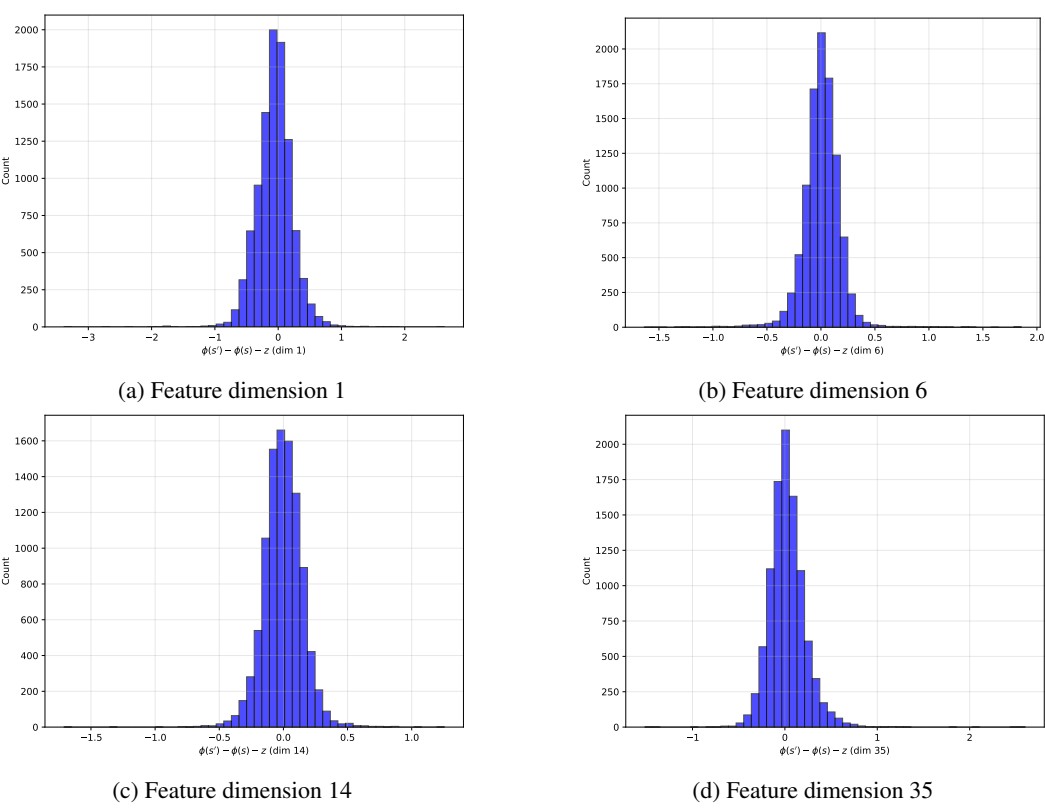

(a) Feature dimension 1

(b) Feature dimension 6

(c) Feature dimension 14

(d) Feature dimension 35

Figure D.6: **Marginal histograms for four randomly sampled feature dimensions in the** *Quadruped* **environment:** the histograms clearly show that Assum. 2(ii) holds across multiple environments.

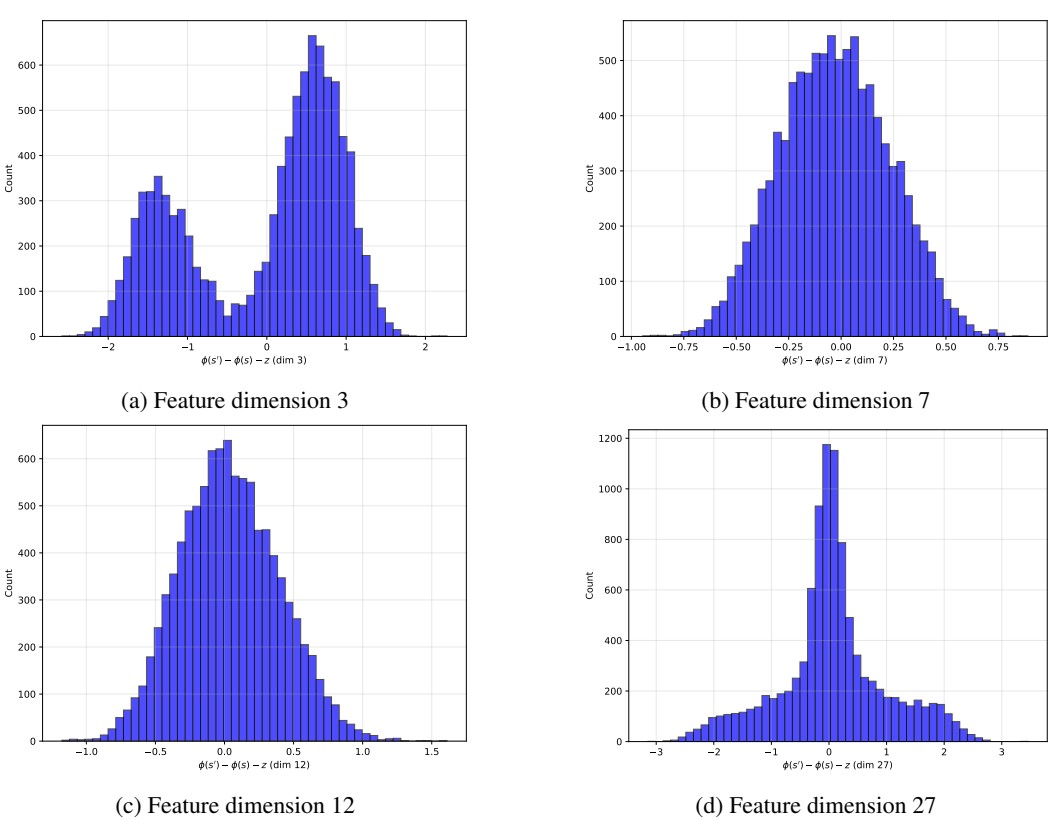

(a) Feature dimension 3

(b) Feature dimension 7

(c) Feature dimension 12

(d) Feature dimension 27

Figure D.7: **Marginal histograms for four randomly sampled feature dimensions in the *Half Cheetah* environment:** the histograms, excluding the single exception of feature dimension 3, show that Assum. 2(ii) holds across multiple environments.

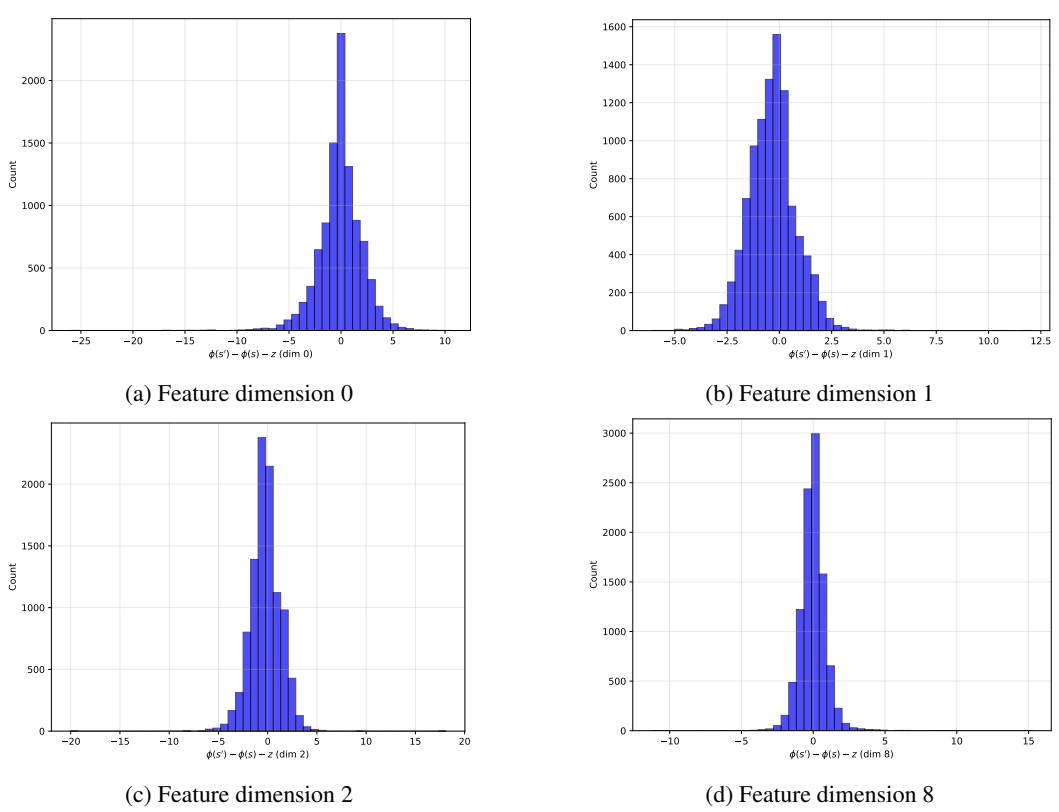

(a) Feature dimension 0

(b) Feature dimension 1

(c) Feature dimension 2

(d) Feature dimension 8

Figure D.8: **Marginal histograms for four randomly sampled feature dimensions in the *Robobin* environment:** the histograms clearly show that Assum. 2(ii) holds across multiple environments.

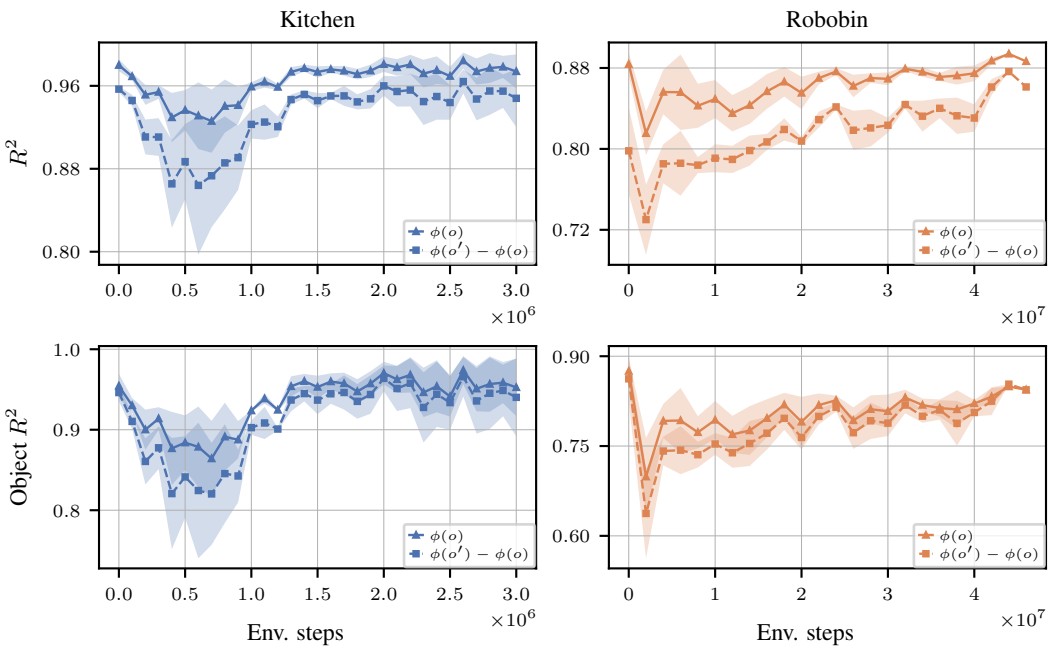

Figure D.9: **CSF identifies both the underlying object states and the full states in the Kitchen and Robobin environments up to a linear transformation.** Object states refer to the object-related subspace of the full ground-truth state. **First row:** Identifiability of both features $\phi(o)$ and feature differences $\phi(o') - \phi(o)$, measured by the $R^2$ score (higher is better); **Second row:** The same metrics computed between the full features and the ground-truth object states. For details, refer to Appx. D.5.

object coordinates only. In Robobin, the object slice (state dimensions 3–8) contains the 3D positions of the two cubes being pushed or picked, excluding the robot hand. In the Kitchen environment, the object slice (dimensions 11–29) covers all manipulated fixtures: the two stove burner knobs, the light switch, the slide and hinge cabinet doors, the microwave door hinge, and the kettle's 7-DoF pose. The strong object $R^2$ values in Fig. D.9 shows that the encoder identifies the manipuland, not just the arm state, up to a linear transformation.

## D.6 GENERALIZATION OF THE ENCODER TO UNSEEN TRAJECTORIES

We evaluate the generalization of the learned encoder representations on trajectories collected by independently trained SAC agents in two state-based and two pixel-based environments: Ant and Half Cheetah resp. Kitchen and Robobin. For each saved SAC snapshot, we feed its ground-truth states into the CSF encoder and report the $R^2$ score between the encoded latents and the ground-truth states. To separate interpolation from extrapolation, we split the SAC states into (i) covered states that fall into the same discretized bins as the states visited by our own method, and (ii) uncovered states that the evaluated CSF agent has not covered in its rollouts. The high $R^2$ scores on the subset covered in Fig. D.10 show the encoder is faithful on already visited states, while performance on the uncovered subset directly measures generalization to expert trajectories that lie outside the representation's training distribution. For the Kitchen environment, there is no existing definition of state coverage; therefore, we only report the $R^2$ score on the unpartitioned rollouts collected by an independently trained SAC agent.

## D.7 DETAILED DEFINITION OF THE COVERAGE METRIC

We follow Zheng et al. (2025) in defining the coverage metric. Let rollouts be indexed by $r = 1, \ldots, R$, each as a list of latent states $s_1^{(r)}, \ldots, s_{T_r}^{(r)}$. For environment $e$, a projection $P_e$ selects specific state components and a discretizer $D_e$ either rounds each selected component to two decimal places or applies $\lfloor \cdot \rfloor$. Define

$$\mathcal{C}_e = \bigcup_{r=1}^{R} \bigcup_{i=1}^{T_r} D_e\big(P_e(s_i^{(r)})\big).$$

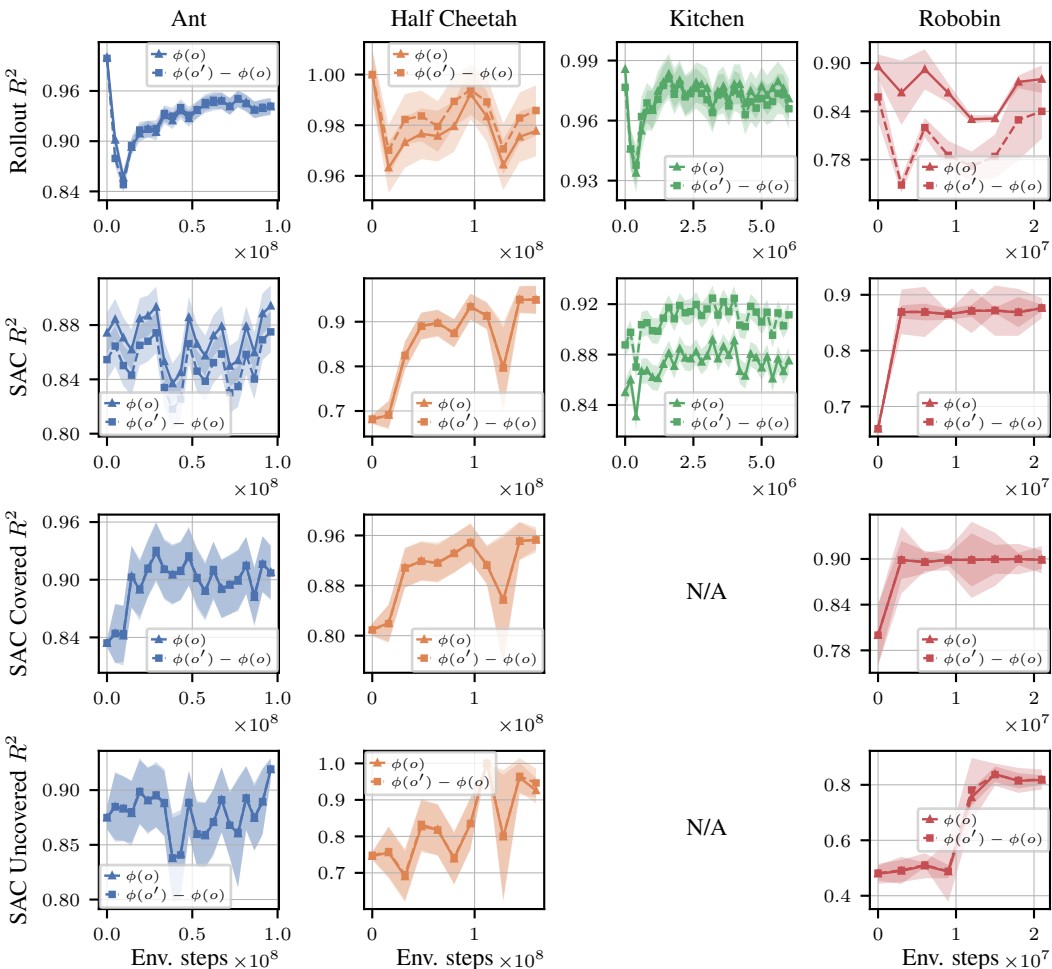

Figure D.10: **CSF shows strong generalization to SAC trajectories that are uncovered by the current CSF agent and remains faithful on the covered trajectories. First row:** The $R^2$ metric between $\phi(o)$ resp. $\phi(o') - \phi(o)$ and the ground-truth state $s$ over rollouts collected by the CSF agent. **Second row:** The $R^2$ metric between the same quantities as in the first row, but over observations that are collected by independently trained SAC agents. **Third row:** Same as the second row, but over only the observations that the CSF agent itself also visited during its evaluation rollouts, measuring in-distribution identifiability. **Fourth row:** Same as the second row, but over the complement states of the third row; i.e., over states collected by the SAC agents that have not been visited by the CSF agent during the rollouts, measuring the generalizability of the encoder's identifiability. Note that the reported $R^2$ scores on the $y-$axes have different scales. For details, refer to Appx. D.6.

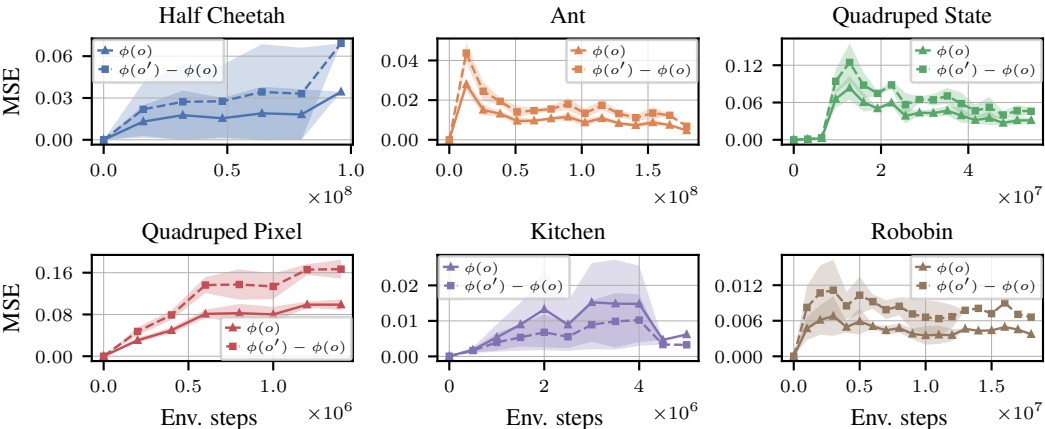

Figure D.11: **CSF accurately reconstructs the ground-truth state from the CSF representation when measured by the MSE metric, not only w.r.t. the $R^2$ score.** For details, refer to Appx. D.8.

State coverage is defined as $\text{Coverage}_e = |\mathcal{C}_e|$.

**Half Cheetah.** $P_e$ keeps the forward state coordinate of the root body of the half-cheetah. $D_e$ applies $\lfloor \cdot \rfloor$. Coverage counts unique floored scalars in $\mathcal{C}_e$.

**Ant.** $P_e$ keeps the planar $(x, y)$ state coordinates of the ant's body. $D_e$ applies $\lfloor \cdot \rfloor$. Coverage counts unique floored 2-dimensional state vectors in $\mathcal{C}_e$.

**Quadruped.** $P_e$ keeps the planar $(x, y)$ state coordinates of the quadruped's body. $D_e$ applies $\lfloor \cdot \rfloor$. Coverage counts unique floored 2-dimensional state vectors in $\mathcal{C}_e$.

**Robobin.** $P_e$ keeps nine state components: the robotic hand and the two objects. $D_e$ rounds each kept component to 2 decimal places. The coverage counts unique 9-dimensional rounded state vectors in $\mathcal{C}_e$.

**Kitchen.** In the Kitchen environment, we define coverage differently, following Zheng et al. (2025). The Kitchen environment contains six tasks: Bottom Burner, Light Switch, Slide Cabinet, Hinge Cabinet, Microwave, and Kettle. The environment logs binary success flags for each task per rollout. The Kitchen task coverage is the number of tasks solved in at least one rollout.

### D.8 MSE Results

In this section, we investigate whether the approximate states recovered from the encoder $\phi$ by a linear probe $W$ are close to the ground-truth states according to the MSE metric. Concretely, we freeze the encoder and fit a linear probe $W$ that predicts the ground-truth simulator state $s$ from the learned representation $\phi(o)$ via $\hat{s} = W\phi(o)$, just like for computing the $R^2$ score. We then compute the MSE between $s$ and $\hat{s}$ using the formula

$$\text{MSE} = \frac{1}{Nd} \sum_{i=1}^{N} \sum_{j=1}^{d} (s_{ij} - \hat{s}_{ij})^2,$$

where $N$ is the number of test samples and $d$ is the state dimensionality. The low errors in Fig. D.11 show that a simple linear decoder can accurately reconstruct the ground-truth state from the CSF representation when measured by the MSE metric, not only w.r.t. the $R^2$ score.

## E  ACRONYMS

**ELBO** evidence lower bound

**CL** Contrastive Learning
**CRL** Causal Representation Learning
**CSF** Contrastive Successor Features

**DGP** data generating process

**ICA** Independent Component Analysis

**MDP** Markov Decision Process
**MI** Mutual Information
**MISL** mutual information skill learning

**POMDP** partially observable Markov Decision Process

**RL** Reinforcement Learning
**RV** random variable

**SSL** self-supervised learning

**USD** unsupervised skill discovery

**vMF** von Mises-Fisher

## F  NOMENCLATURE

$CDR^2$ coverage-dependent coefficient of determination

$R^2$ coefficient of determination
$\mathcal{S}$ hypersphere

