# OpenReview forum: "Skill Learning via Policy Diversity Yields Identifiable Representations for Reinforcement Learning"
_ICLR.cc/2026/Conference — ICLR 2026 Poster_

### Official Review · Reviewer_YV3A · 2025-10-30

**Soundness:** 2
**Presentation:** 2
**Contribution:** 3
**Rating:** 4
**Confidence:** 2

**Summary:**

This paper offers an explanation for the effectiveness of skill-learning methods based on mutual information (MISL). The authors provide both theoretical and experimental analyses of how the representations learned by a representative approach (CRL) of MISL relate to the true latent factors of the environment. Their results show that, under certain assumptions, CRL implicitly learns representations that are linearly related to the environment’s underlying latent state.

I believe the paper touches an important topic and provides an interesting and novel analysis on the representations leaned by skill learning methods. However, the writing could be improved and the analysis could be strengthened to support the claims.

Writing:
- The introduction could be made easier to follow. Some parts would fit better in the related work section, and long citation lists should be avoided. The connection to causality is not entirely clear, and lines 81–89 do not seem directly related to the paper’s main motivation.
- The explicit definition of identifiability should appear earlier in the manuscript to help readers grasp the theoretical framework. The discussion in the introduction is not sufficiently explicit.
- I recommend separating the related work from the background section to improve readability and structure.
- Line 156: This paragraph lacks clarity and could benefit from rephrasing or additional context.
- Definition 1 (line 171): The proposed definition seems closer to “distinguishability” than “diversity.” Two skills may produce very similar trajectories yet still represent different behaviors. The relationship between this definition and the notion of diversity used in Assumption 1 should be clarified.
- The statement “Each pair of consecutive states corresponds to one skill” introduces a strong and limiting assumption. This is unlikely to hold in many environments. For instance, an agent might need to traverse a corridor before exploring an area, or operate in an MDP with discrete actions and deterministic transitions. A more thorough discussion of such edge cases would strengthen the paper. Furthermore, the connection to the statement line 293 is unclear.
- In Figure 2 (right), the direction of the arrows is ambiguous and should be clarified.
- First paragraph of Section 3.2. I do not catch why z and feature representations could be antipodal by maximizing their dot product.
- In Section 3, the example at line 241 is particularly helpful and could be emphasized earlier in the section. However, the statement "to distinguish data under different distribution shifts or interventions" is not "intuitive".

Experiments:
- In Figure 3, using features or feature differences do not induce meaningful difference in identifiability. This should be discussed. Overall, it lacks a baseline that gives indications of what is a high identifiability.
- The authors should formally define state coverage in the manuscript.
- The authors rightly note that the R2 score is influenced by state coverage, but the CDR² curve remains difficult to interpret and, in practice, seems to primarily reflect coverage rather than the main contribution of this work.To clarify this point, the authors should compute the R2 score over a set of presampled agent positions, either manually defined or extracted from trajectories of trained DRL agents. Optionally, they could separately evaluate the R2 for covered and uncovered states to analyze potential generalization properties of the learned representations.
- From Figure D1, it seems relatively easy to achieve high identifiability in these experiments. The metric starts very high, drops sharply when learning begins, and then rises again. This pattern raises concerns that learning state-related representations might be too simple in these environments, and therefore not representative of more complex scenarios. Some environments include objects, i.e. parts of the true state that are relatively difficult to manipulate and represent. To strengthen the paper’s validity, the authors should include a correlation coefficient that relates to only these parts of the state space and discuss the result with respect to the ability of the agent to reach these states.

**Strengths:**

The paper makes an original contribution, supported by both theoretical and experiment results.

**Weaknesses:**

The writing should be improved and the analysis should be strengthened.

**Questions:**

Please, see above.

---

> ### Author Response · Authors · 2025-11-24
>
> We thank the Reviewer for deeming our submission to be an original contribution to the field and also for the feedback. We appreciate the detailed suggestions for improvement; we believe that the revised paper is stronger because of these revisions. We provide a summary of our response (and detail each below):
>
> **Summary:**
> - **We have clarified Def. 1**
> - **We have clarified Fig. 2 and the implications of MI formulations**
> - **We added more experiments to demonstrate generalization and good performance on object manipulation tasks (Appx. D.5 and D.6).**
> - **We have restructured the introduction and background (which now includes a detailed discussion of identifiability), and now include a separate related works section.**
> - **We added a detailed explanation on**  CDR².
>
> ### Related works and background
> Following the reviewer's recommendation, we have restructured the introduction and background, and now include a separate section for related work. Furthermore, we formalize the notion of identifiability in Sec. 3.1.
>
>
> ### Definition 1
> Thank you for pointing out this discrepancy\! We originally chose diversity to follow the convention of “diversity” assumptions in the identifiability literature. We have realized that the condition on the skills forming an affine generator system was not part of our Def. 1, only that of Assm. 1\. For emphasis, we added the requirement on forming an affine generator system.
>
> ### Clarification: skills and consecutive states
> Thank you for pointing out that our formulation was unclear in the informal assumption. Now we have restructured both Def. 1 and Assms. 1 (the informal) and 2 (the formal) to clarify that what we meant is that an ideal discriminator can uniquely map $(s'-s) \\mapsto z$, i.e., for each pair of consecutive states there is a single corresponding $z\_k$, but for multiple state pairs it can be the same. We also added a clarification to Assm. 2 as a footnote.
>
> ### Fig. 3 and identifiability scores
> That’s a correct observation. Since both the differences and the particular features are extracted from the same model, by both Props 1 and 3, we know that both quantities are identifiable. Thus, we do not expect large deviations. We would like to contextualize our results: Since the $R^2$ score is at most 1, and comparing to the identifiability literature, scores above 0.8 can be considered as high.
>
> Furthermore, to verify the premise of identifying the representation (i.e., generalization), we conduct further experiments (Appx. D.6 and Fig. D.10) to demonstrate that the representation learned by CSF identifies the ground-truth factors even if the trajectories were collected by another policy (so to speak, “offline”). Concretely, we train independent SAC agents on the same environments, collect states from their replay buffers, and feed these states through our frozen CSF encoder to compute $R^2$ between the decoded features and the ground-truth states. We then split these SAC states into
> - "covered" states (bins that the currently evaluated CSF agent has visited in its rollouts), and
> - "uncovered" states (the rest of the bins),
> and repeat the calculation of the metrics.
>
> On the covered (in-distribution) states, the $R^2$ scores show minimal degradation (in all four environments, they are around or above 0.9), whereas on the uncovered (out-of-distribution) states, the $R^2$ scores are almost the same, except in Robobin, where they drop from around 0.9 to just above 0.8, which is still considered a high score.
>
> ### Elaborating on  CDR²
>
> Thank you for pointing out that the definition and interpretation of $CDR^2$ were not sufficiently explicit. We have expanded App. C.3 to make the construction and role of the metric clearer.
>
> Concretely, we now spell out that $CDR^2$ is defined as the harmonic mean between (i) the relative state coverage $\\tilde C \\in \[0,1\]$ (normalized by the highest attainable coverage) and (ii) the standard $R^2 \\in \[0,1\]$:
> $$
> CDR^2 \= \\frac{2\\tilde CR^2}{\\tilde C \+ R^2}.
> $$
> The harmonic mean is dominated by the smaller of its two arguments. Thus:
> - if $R^2$ is high but coverage is poor, $CDR^2 \\approx 2\\tilde C$ and remains small;
> - if coverage is high but the representation does not linearly identify the state (low $R^2$), then $CDR^2 \\approx 2R^2$ and again remains small.
>
> We agree with your observation that in some of our curves $CDR^2$ visually tracks coverage quite closely. This is largely because in these runs $R^2$ saturates very early to values $\\gtrsim 0.9$ (cf. Figs. D.1–D.2), so exploration becomes the bottleneck. In that regime, by construction, the harmonic mean is dominated by relative coverage. We now state this explicitly in the text to avoid the impression that $CDR^2$ is meant to completely replace the individual curves: in the main paper, we use it only as a compact summary metric, while the separate $R^2$ and coverage plots are kept in App. D to disentangle the two contributions.

---

> > ### Author Response · Authors · 2025-11-24
> >
> > ### Correlation of object-related parts of the state space
> >
> > We agree that assessing identifiability only on the full simulator state can obscure whether the more challenging, object-related factors are actually recovered. Following your suggestion, we have added an "object-only" identifiability metric in App. D.5 (summarized in Fig. D.9).
> >
> > For the two manipulation environments, we explicitly define an "object slice" of the ground-truth state:
> > - RoboBin: dimensions 3–8, corresponding to the 3D positions of the two cubes being pushed or picked, excluding the robot hand.
> > - Kitchen: dimensions 11–29, corresponding to the manipulable fixtures (stove knobs, light switch, sliding and hinge cabinets, microwave door, and the kettle’s 7-DoF pose).
> >
> > We then train the same linear probe as in the main $R^2$ metric, but now only predict these object coordinates from the learned representation, and report an “object $R^2$” alongside the full-state $R^2$. As shown in Fig. D.9, the object $R^2$ quickly rises and closely tracks the full-state $R^2$ in both Kitchen and RoboBin, reaching values around $\\sim 0.84$–$0.95$ once training has stabilized.
> >
> > These results show that CSF not only encodes the proprioceptive state of the agent but also identifies the more difficult object-centric degrees of freedom up to a linear transformation. We now explicitly emphasize this point in Appx. D.5.
> >
> > ### Figure 2 and  antipodal z and features
> > Thanks for your question\! We realize that we have made an unclear statement. The antipodality issue can only arise if both the skill and the embedding are multiplied by minus one. We have realized that both our formulation and Figure 2 were misleading. We have now corrected our statement in both the caption and Sec. 3.2.
> >
> >
> > ### Connection to causality
> > We have clarified this connection. Namely, the central role of interventions in causality (which corresponds to actions) and the application of causal representation learning to robotics scenarios (Sec. 2.3).

---

> > > ### Comment · Reviewer_YV3A · 2025-11-27
> > > **Response to authors**
> > >
> > > I thanks the authors for their answer, clarifications and additional experiments. Please, for next revisions, hightlight the changes with a color.
> > >
> > > - Regarding skills and consecutive states, there is a misunderstanding. It is unrealistic, in many scenarios, to assume that a pair of states can uniquely be mapped to one skill.
> > >
> > > I still have concerns and questions regarding the experiments:
> > >
> > > - 0.8-0.9 may be a high absolute identifiability score, but this may not be a high score relative to the neural networks and considered tasks. Therefore, it is necessary to provide comparison baselines. This should at least include a random network (I am not 100% sure that this corresponds to the far left value shown in each plot) and intermediate representations of the SAC algorithm. For a stronger baseline: I suggest the latent space of a simple auto encoder trained on trajectories. A positive comparison with a VAE is unnecessary given the scope of the paper, but the demonstration of a positive advantage over a VAE would increase the impact of the paper.
> > > - In Figure D.10, some results are provided for only one pixel-based environment. I would like to see the results for at least one other pixel-based environment. As acknowledged by the authors, the encoder may just learn the identity transform map in state-based environments: this makes it hard to appreciate the results. An alternative way may also be to first entangle the states by processing them with a fixed random neural network, before using them during training.
> > > - In Figure D.10, could the authors comment on the temporal variability of the identifiability score ? The variability is surprisingly consistent over random seeds.
> > > - Finally, are "covered"  and "Rollout" states defined based on states visited by an agent at the end of training, are at a given evaluated timestep ?

---

> > > > ### Author Response · Authors · 2025-11-30
> > > >
> > > > Thank you for your further feedback and engagement in the rebuttal! We would like to clarify our experimental setup and address your concerns below.
> > > >
> > > > ### **Assumption on skills and consecutive states**
> > > >
> > > > We apologize for the remaining confusion. We add an additional formulation, which hopefully clarifies the assumptions and shows that it is not unrealistic.
> > > >
> > > > **What our assumption *does* mean**
> > > >
> > > > - Our assumption means that each state pair $(s,s')$ can be mapped to a skill via an ideal discriminator.
> > > > - Alternatively: if we treat this as a classification problem, then we can say that for each state pair $(s,s')$ there is a unique “most probable” class (skill), **but** $(s,s')$ can “belong” (to some extent) to multiple classes (there will be a “most probable class” though). That is, the setup is akin to a Gaussian Mixture Model setup.
> > > >
> > > > **What our assumption does NOT mean**
> > > >
> > > > - We **do not** assume a one‑to‑one correspondence between skills and state pairs in the sense that “each skill corresponds to a unique $(s,s')$”.
> > > > - We **do not** assume that each state pair can only belong to one skill
> > > > - We **do not** assume that only trajectories generated by a skill-conditioned policy for a single given $z\_i$ can visit $(s,s’)$. \[That is, multiple skills can imply trajectories that have $(s,s’)$ as consecutive state pairs in them.\]
> > > >
> > > > We have now rephrased this assumption more carefully (and added an explicit remark) to avoid suggesting a one‑to‑one correspondence between skills and individual transitions.
> > > >
> > > > ---
> > > >
> > > > ### **On the magnitude of the identifiability scores and baselines**
> > > >
> > > > **Random network baseline.**
> > > >
> > > > The leftmost point in Figs. 3, D.1 and D.2 indeed corresponds to a completely **randomly initialized encoder and policy**, i.e. exactly the “random network” baseline the reviewer suggests. We now state this explicitly in the experimental details in Appendix C.3.
> > > >
> > > > As we also discuss in there, a random encoder combined with a *non‑exploring* random policy can already achieve a deceptively high $R^2$ when evaluated **only on the very small region of state space that is initially visited**. This is precisely why we introduce the coverage‑dependent score
> > > > $$ \mathrm{CDR}^2 \= \\frac{2 \\tilde C R^2}{\\tilde C \+ R^2}, $$
> > > >  which is near zero at initialization because the relative coverage $\\tilde C$ is essentially zero, even though $R^2$ can already be high on the visited states. We emphasize this effect in the updated text and figure captions.
> > > >
> > > > In contrast, at convergence we obtain both **high coverage and high $R^2$**, leading to $\\mathrm{CDR}^2 \\approx 0.8$–0.9, which is very close to the maximal attainable value and substantially above the random‑network baseline, whose $\\mathrm{CDR}^2$ remains near zero.
> > > >
> > > > **SAC “intermediate representation” baseline.**
> > > > SAC does not contain a dedicated trajectory encoder analogous to CSF’s $\\phi(o)$; its function approximators are trained directly to predict extrinsic returns. Adding a *new* encoder on top of SAC for the sake of comparison would defeat the purpose of comparing to “plain SAC” and introduce substantial design choices (architecture, training objective, regularization) that would need their own careful tuning and analysis.
> > > >
> > > > Instead, we use SAC in App. D.6 exactly as a **distribution shift**: we train SAC agents independently and evaluate how well the *CSF encoder* identifies the ground‑truth state on these SAC trajectories, including states that CSF never visited during pre‑training. This directly probes the robustness and generalization of the CSF representation relative to a strong model‑free RL baseline, while keeping the representation architecture fixed.

---

> > > > > ### Author Response · Authors · 2025-11-30
> > > > >
> > > > > **Autoencoder / VAE baselines.**
> > > > > We agree that comparing against purely reconstruction‑based encoders (autoencoders, VAEs) would be interesting in its own right. However, such baselines are somewhat orthogonal to the main question of the paper—whether **skill‑learning with an InfoNCE‑style objective leads to identifiable representations**. Furthermore, there are identifiability results for (V)AE-based architectures \[1-4\], which would imply that any comparison \- given our high identifiability scores \- could only establish that (V)AE-based models can *also* identify the ground-truth factors, which, we believe, was not the intent of the reviewer’s suggestion.
> > > > >
> > > > > We would like to emphasize that our experiments provide strong evidence for our identifiability theory:
> > > > >
> > > > > * (i) **Absolute identifiability numbers** ($R^2$ and MSE) that are close to 1 in both state‑ and pixel‑based environments (App. D.1–D.2, D.8), and
> > > > >
> > > > > * (ii) **Relative comparisons** across design choices (skill diversity, latent dimensionality) that strongly support the identifiability interpretation (Figs. 4–5, D.3–D.4).
> > > > >
> > > > > \[1\] Hyvärinen, A., Hurri, J., & Hoyer, P. O. (2001). Independent component analysis. In *Natural Image Statistics: A Probabilistic Approach to Early Computational Vision* (pp. 151-175). London: Springer London.
> > > > >
> > > > > \[2\] Khemakhem, I., Kingma, D., Monti, R., & Hyvarinen, A. (2020, June). Variational autoencoders and nonlinear ica: A unifying framework. In *International conference on artificial intelligence and statistics* (pp. 2207-2217). PMLR.
> > > > >
> > > > > \[3\] Hyvarinen, A., & Morioka, H. (2017, April). Nonlinear ICA of temporally dependent stationary sources. In *Artificial intelligence and statistics* (pp. 460-469). PMLR.
> > > > >
> > > > > \[4\] Reizinger, P., Gresele, L., Brady, J., Von Kügelgen, J., Zietlow, D., Schölkopf, B., ... & Besserve, M. (2022). Embrace the gap: VAEs perform independent mechanism analysis. *Advances in Neural Information Processing Systems*, *35*, 12040-12057.
> > > > >
> > > > > ---
> > > > >
> > > > > ### **Pixel‑based environments and “entangled” observations**
> > > > >
> > > > > **Number of pixel‑based environments in Fig. D.10.**
> > > > > Fig. D.10 in the current version reports generalization results on **two pixel‑based environments**, Kitchen and RoboBin, in addition to the two state‑based MuJoCo tasks. For RoboBin, we also split the SAC trajectories into “covered” and “uncovered” subsets (see below). For Kitchen, this split is not defined because there is no notion of state‑space coverage in terms of spatial bins, so we report a single SAC curve for that environment. We have made this more explicit in the caption and in App. D.6.
> > > > >
> > > > > **“Entangling” states with a fixed random network.**
> > > > >  We appreciate the suggestion. Conceptually, the pixel‑based environments already implement precisely such an “entangling” generator  $ g : s \\mapsto o, $
> > > > >  which is a complex, high‑dimensional, nonlinear rendering function. Training CSF on pixels and linearly decoding the simulator state from $\\phi(o)$ therefore already tests whether we can “invert” a fixed but unknown entangling map $g$, which is exactly the setting assumed in Assumption 2(v).
> > > > >
> > > > > Composing the simulator state with an additional fixed random neural network before feeding it to the encoder would amount to replacing $g$ by $\\tilde g \= h \\circ g$ for some injective $h$. Under Assumption 2(v), this does not change the identifiability guarantees (they are stated up to an unknown injective generator), so we would expect qualitatively identical conclusions. To keep the experimental section focused, we therefore kept the physically meaningful pixel‑rendering settings rather than adding a second, synthetic entangling layer.
> > > > >
> > > > > ---
> > > > >
> > > > > ### **Temporal variability and error bars in Fig. D.10**
> > > > >
> > > > > Thank you for pointing out the strangely consistent variability. Upon re‑checking, we discovered a copy‑and‑paste error when aggregating the standard deviations for the first two columns of rows 2–4 in Fig. D.10. This affected only the **visualization of the error bars** in this figure, not the underlying mean values.
> > > > >
> > > > > We have corrected this bug in the revised manuscript; the updated figure now shows small but non‑zero variability across seeds, as expected: early in training, the encoder is still changing, and the curves exhibit more spread, while after convergence, the $R^2$ values are close to saturation and vary only slightly across seeds. We also added a short remark about this in App. D.6.

---

> > > > > > ### Author Response · Authors · 2025-11-30
> > > > > >
> > > > > > ### **Definition of “Rollout”, “Covered” and “Uncovered” states**
> > > > > >
> > > > > > We realize this was not explained clearly enough and have clarified it in App. C.3 and D.6. The definitions are:
> > > > > >
> > > > > > * **“Rollout”** (first row of Fig. D.10): at each evaluation checkpoint during CSF pre‑training, we run evaluation rollouts with the **current CSF policy** and collect the states visited at that checkpoint. The “Rollout” $R^2$ curves are computed on this set of states and therefore reflect in‑distribution identifiability as training progresses.
> > > > > >
> > > > > > * **“SAC” (second row):** for each CSF checkpoint, we evaluate the *same* encoder on a fixed set of trajectories generated by independently trained SAC agents in the same environment. This probes how well the current CSF representation identifies the ground‑truth state under a different data‑generating policy.
> > > > > >
> > > > > > * **“SAC covered/uncovered” (third and fourth rows):** for a given CSF checkpoint, we discretize the state space as in App. C.3 and define
> > > > > >
> > > > > >   * **covered SAC states** as those whose discretized coordinates fall into bins that have been visited by the **currently evaluated** CSF agent, and
> > > > > >
> > > > > >   * **uncovered SAC states** as those falling into bins that the CSF agent has not visited yet.
> > > > > >
> > > > > > * We then compute $R^2$ separately on these two subsets. This lets us distinguish between **interpolation** (SAC‑covered states) and **extrapolation/generalization** (SAC‑uncovered states) for the same encoder and time step.
> > > > > >
> > > > > > We hope that this clarifies the construction and interpretation of the curves in Fig. D.10.

---

### Official Review · Reviewer_vUpn · 2025-10-31

**Soundness:** 2
**Presentation:** 3
**Contribution:** 2
**Rating:** 4
**Confidence:** 4

**Summary:**

This paper investigates identifiability in mutual information skill learning and contrastive successor features (CSF). The authors provide some evidence for the ability of CSF representations to recover the ground truth state used to produce the CSF because of the inner product parameterization of the loss. Further, the paper investigates the theoretical merit of different instantiations of the MISL objective, whether to use $\phi(s)^Tz$, $[\phi(s) - \phi(s')]^T$, or $[\phi(s_0) -\phi(s)]^Tz$. Finally, the authors provide some identifiability results on common RL tasks in the Deep Mind control environments.

**Strengths:**

This paper is well motivated and well written. Its objectives are clear and, to my knowledge, provide the first analysis of ground truth identifiability using mutual information skill learning (MISL) losses. The paper introduces the notion that a set of diverse skills and an inner product parameterization are necessary for learning a robust representation that provably recovers the ground truth state.

**Weaknesses:**

There are several weaknesses that exist are present in the paper that must be addressed.

## Major Weaknesses
1. **Reality of assumptions**: It is not clear that the assumptions made in the paper are representative of reality. Namely, is it common that "each state difference is equiprobable"? What is the support for this claim?
2. **Transitions are typically not skills**: The authors also assume that "each pair of consecutive states is a skill". I believe that this is not a typical definition of skills. Skills are often defined on a longer time horizon and so single-step transitions are not wholly descriptive of what is traditionally called a skill. Further, there is literature that demonstrates that the controllable state (i.e., features that are affected by actions) are not identifiable using single-step transitions (see [1]). With that in mind, i believe it is not reasonable to define skills as (s,s') pairs.
3. **Lacking explanation for necessity of inner product parameterization**: The authors repeatedly claim that the inner product parameterization is important for identifiability, but do not share any evidence towards this end. Is there a specific result that they found that would indicate this? Or is this notion sourced from previous literature?
4. It is not clear to me why $I(s; z)$ or $I(s_0, s; z)$ are reasonable alternatives to $I(s, s'; z)$ as the former two information quantities would imply things about the visitation distributions alone. Is there a valid reason to consider these as alternatives to $I(s, s'; z)$?

### Minor Weaknesses

1. The authors claim that  $\phi(s) - \phi(s')$ is typically a learning target (line 161). Can the authors point to a method that uses such a construction? Are the authors are referring to metric-based methods like value implicit pre-training where $\|phi(s) - \phi(g) \| $ is used?
2. It believe it would be good to see actual MSE on extracted states as opposed to the correlation coefficients. In my experience, it is very possible to extract the ground truth state from encoded images in DM Control.
### Nitpicks
Line 267 has colon at end of line

**Questions:**

Please refer to the weaknesses section for my questions about the work.

---

> ### Author Response · Authors · 2025-11-24
>
> We appreciate the Reviewer’s acknowledgement of our work being well-motivated and well-written and also the detailed suggestions for improvement; we believe that the revised paper is stronger because of these revisions, which we summarize as follows (see our detailed responses below).
>
> **Summary:**
> - We have clarified the notion of diversity and the state pair/skill correspondence in Secs. 2.2 and 3.1.
> - We provided evidence from the literature on the importance of the inner product parametrization
> - We have clarified that we considered different parametrizations of the mutual information.
> - We have clarified the role of feature differences.
>
>  ### is it common that "each state difference is equiprobable"?
> It is a common technical assumption in nonlinear ICA theory, especially in contrastive methods (see the theoretical analysis of the InfoNCE loss by ZImmermann et al., 2021 or its recent extension by Rusak et al., 2025), and it is reflected by empirical evidence in Fig. 2, right of Zhang et al. 2024.
>
>  ### clarification: skills and consecutive states
> Thank you for pointing out that our formulation was unclear in the informal assumption. Now we have restructured both Def. 1 and Assms. 1 (the informal) and 2 (the formal) to clarify that what we meant is that an ideal discriminator can uniquely map $(s'-s) \mapsto z$, i.e., for each pair of consecutive states there is a single corresponding $z_k$, but for multiple state pairs it can be the same. We also added a clarification to Assm. 2 as a footnote.
> Thank you for the reference. We hope that our clarification above makes clear that, since skills are not defined as state pairs, the cited non-identifiability concern does not apply.
>
>
>
>  ### inner product parametrization
> Sorry for the confusion! In L180-182, we provide references to the ICA literature, demonstrating that **one of the key components to identifiability guarantees is the inner product parametrization** (Hyvarinen et al., 2019, Zimmermann et al., 2021, Rusak et al., 2025, Reizinger et al., 2025).
>
> Furthermore, **Zhang et al. (2024) provide experimental evidence** that, in MISL, too, inner-product parametrization is key for good state coverage (c.f. Their Fig. 3, right), which is a prerequisite for identifiability (if some states are not covered, then the corresponding ground-truth factors cannot be identified).
>
>  ### It is not clear to me why $I(s; z)$ or $I(s_0, s; z)$ are reasonable alternatives to $I(s, s'; z)$ as the former two information quantities would imply things about the visitation distributions alone. Is there a valid reason to consider these as alternatives to $I(s, s'; z)$?
>
> That’s a good question. What we **intended with the discussion of the parametrization of mutual information was to provide an insight into why $I(s, s'; z)$ is a better choice**. We do not say that  $I(s; z)$ or $I(s_0, s; z)$ are reasonable alternatives; we only consider those as other USD methods in the literature use  $I(s; z)$ or $I(s_0, s; z)$ - as it was reviewed, e.g., by Laskin et al., 2021 (Table 2).
>
>  ### The authors claim that $\phi(s) - \phi(s')$ is typically a learning target (line 161). Can the authors point to a method that uses such a construction? Are the authors are referring to metric-based methods like value implicit pre-training where $|phi(s) - \phi(g) | $ is used?
> Thanks for the question! **What we intended to say was that many USD methods rely on a $\phi(s) - \phi(s')$ parametrization in their objective functions, including METRA and CSF**---where this comes from the InfoNCE loss formulation, which also takes the difference between two embeddings (in that case, due to the unit-norm constraint on the embeddings, the equivalent inner-product parametrization is used). We hope this clarifies our point, which we made clearer in our updated submission.
>
>  ### Correlation vs. MSE
> We agree with the reviewer that, in addition to the $R^2$ and $CDR^2$ metrics, the MSE between the states extracted from the encoder representation and the ground-truth ones is another valuable metric to report. **We provide results for all environments in Section D.8**. In short, we find that **CSF accurately reconstructs the ground-truth states, as measured by the MSE metric.**

---

### Official Review · Reviewer_o5Lj · 2025-11-01

**Soundness:** 3
**Presentation:** 3
**Contribution:** 3
**Rating:** 8
**Confidence:** 4

**Summary:**

This work aims demonstrate that the causal features of an environment can be identified using contrastive successor features from up to a linear transformation. This forms an identifiability guarantee for unsupervised RL in the POMDP setting, so that the true states are derived from observations. Then, the work demonstrates empirically that the ground-truth features in several domains are recovered using CSF.

**Strengths:**

This work applies an elegant description of identificabiltiy in a novel context.

The theoretical framework is well articulated and provides clear reasons advantages.

The empirical results are sufficient to support the theoretical claims

**Weaknesses:**

The empirical results are somewhat limited in scope, considering the extension to the POMDP setting

The work is not particularly self contained, in that many of the claims are fully described in the appendix.

CSF is not the most representative MISL algorithm because it detaches teh representation learning from the policy learning more than most methods.

**Questions:**

It might be easier to read a slightly less aggressive citation style (the green and red boxes inserted by the citation links make the make it hard to parse the introduction).

The second paragraph of the introduction is unclear: How does the natural connection to causality blur the borders between these fields? Interconnection does not produce a distinction between decisionmaking with scalar rewards (RL) and cause-effect models.

Traditionally, unsupervised skill discovery is in the setting of fully observed MDPs rather than partially observed, so the focus on identificability results is somewhat jarring. This focus could be introduced a bit earlier rather that in the description of ICA.

Does the definition of diversity (definition 1) really capture the meaning? It seems like diversity should also cover some description of the breadth of skills, but in this case any pair of skill parameters is sufficient for diversity. Maybe a term such as distinguishable is more appropriate.

While CSF is a meaningful algorithm, is it really "prototypical" (l183) of MISL methods? In many cases it is closer to a representation learning algorithmm with a skill parameter added to it, rather than a MISL algorithm which often has the representation learning component built into the skill learning.

It makes sense that the assumptions are required to guarantee identifiability, but in order to make the claim that identificability is robust to assumption violations in practice it does not seem sufficient to simply claim success in other fields, since RL is a significant different setting, especially since the agent is part of the data generating process. Is there some stronger evidnece to suggest robustness to assumption violations, especially theoretically.

THe fact that Empirical observations show that the fatures of consecutive observations are close is not entirely convincing universally, since in many cases with locomotion tasks that these algorithms are tested on, the observations are often visually close together as well (an ant moving around will have similar appearances), so this closeness could just be a consequence of this.

While it is certainly the case that I(s,s';z) is more stable to optimize, it seems like this state difference may not always be the best choice for representing a variety of different policies, since identifying policies from their state difference means that it can be hard to distinguish when policies should start changing their direction.

The experimental domains are sufficeint for demonstrating that the underlying state is discovered, though they are certainly not complete. In particular these environments focus on motion planning with various morphologies, not on the manipulation of different eleemtns in the domain, of whose underlying state might be much harder to describe, especially when randomly initialized.

Is the true underlying state really the state of the joints in mujoco, since to some extent this also is a layer of abstraction abouve the length of the joints or the shape of the agent.

---

> ### Author Response · Authors · 2025-11-24
>
> We thank the reviewer for their acknowledgement of our work as well-articulated and for providing an elegant description and their detailed feedback, which improved the quality of our submission. We made the following changes to our submission (we detail our responses below):
>
> **Summary:**
> - We clarified the second paragraph of the Introduction.
> - We emphasized the nuance of full vs partial observability in the MDP earlier, in the introduction, and highlighted that the key is that the states are not always observable (even though the skills are).
> - We added new experimental results to demonstrate the generalization of CSF and that it can learn good representations in object manipulation, not only in locomotion tasks (Appx. D5. and D.6)
>
>  ### How does the natural connection to causality blur the borders between these fields?
> That’s a great question! A wide range of intervention types corresponds to actions in the reinforcement learning terminology. On the other hand, identifiability is also a central question in causal representation learning. Also, RL methods often learn a representation on top of which they deploy their policy. If we interpret the RL agent as the entity that provides us interventional data, then we can apply the framework of identifiable (causal) representation learning. We clarified the second paragraph of the Introduction.
>
>  ### Clarification on observability: POMDP/MDP
>
> We agree with the reviewer that many USD methods are proposed for fully observed MDPs. However, when these methods are deployed in pixel-based environments, they need to operate under partial observability. We have added emphasis and mentioned this in the introduction (L076).
> Importantly, although the skills are observed, the non-triviality of our result comes from the fact that the ground-truth states are not necessarily observed (e.g., when learning from pixels). Thus, it is not guaranteed to recover the ground-truth states.
>
>  ### Diversity definition clarification
> Thank you for pointing out this discrepancy! We originally chose diversity to follow the convention of “diversity” assumptions in the identifiability literature - methods include TCL, GCL, ICE-BeeM, IIA, IEM; for an overview, refern to Hyvarnien et al., 2019 or Reizinger et al., 2025. We have realized that the condition on the skills forming an affine generator system was not part of our Def. 1, only that of Assm. 1. For emphasis, we added the requirement on forming an affine generator system.
>
>
>  ### While CSF is a meaningful algorithm, is it really "prototypical" (l183) of MISL methods?
>
> To the best of our knowledge, most of the recently proposed SOTA methods share the same underlying principles---even though they are sometimes implemented in (slightly) different ways. These components are:
> - The use of a self-supervised (often InfoNCE-based or -inspired) representation learning objective
> - An inner-product parametrization
> - And, by definition, a mixture policy
>
>
>  ### Mutual information parametrization and using state differences
> Yes, we agree that this parametrization might not work well universally. We have opted for this based on the suggestion of Zheng et al. (2024), who found that a difference-based parametrization is more stable.
>
>
>  ### Experimental diversity: motion planning and object manipulation
> Although the Ant, HalfCheetah, and Quadruped environments focus on locomotion, both Kitchen and RoboBin include object manipulations. As prompted by Reviewer YV3A, we conducted additional investigations to **demonstrate that CSF can recover the ground-truth factors specific to manipulable objects (Appx. D.5 and Figure D.9),** providing further evidence that CSF can recover the ground-truth states across a somewhat diverse range of environments.
>
>  ### Choice of state and the problem of abstraction
> Yes, that’s a correct observation! To **discuss representational identifiability, one must choose a level of abstraction to evaluate identifiability performance.** The field of causal abstraction studies this question. However, we are not aware of identifiable representation learning methods that can explicitly control the level of abstraction; thus, we follow the convention of the identifiability literature and evaluate by assessing the recovery of the ground-truth (in this case, the simulator) state.

---

> > ### Author Response · Authors · 2025-11-24
> >
> > ### Robustness to assumption violations.
> >
> > We agree that our formal guarantees are stated under idealized assumptions (Assumption 2) and that it is important to argue why the mechanism should remain useful when these assumptions are only approximately satisfied in RL. Here, we make this connection more explicit.
> >
> > **Several aspects of our experimental setups deliberately violate Assumption 2, yet CSF still yields high identifiability scores**:
> > - In the Kitchen environment, multiple coordinates of the simulator state correspond to discrete events (e.g., light switch on/off, cabinet doors open/closed, contact/no‑contact with an object). This induces strongly multi‑modal marginals and contact discontinuities that do not fit the single von Mises–Fisher component assumed in Assumption 2(ii). Nevertheless, CSF recovers both the full simulator state and the object-only subspace with a high $R^2$ score, showing that the linear identifiability mechanism is robust to such deviations.
> > - In the pixel-based environments (Quadruped Pixel, Kitchen, RoboBin), the rendering function $g: s \mapsto o$ is not injective in practice because of occlusions and finite image resolution; thus, different simulator states can produce indistinguishable observations. This violates Assumption 2(v). Yet, linear decoders trained on our learned features still reconstruct both the full state and object coordinates from pixels with a high $R^2$ score.
> > - Appendix D.4 shows that, even in our state-based HalfCheetah environment, some latent dimensions of the learned feature differences are bimodal rather than Gaussian/vMF. Despite this violation of Assumption 2(ii), the $R^2$ and $CDR^2$ scores remain high throughout training.
> > - Finally, our freshly added Appendix D.6 evaluates the same encoder on trajectories generated by independently trained SAC agents, both on states that our CSF agent has visited and on states that are never visited during CSF pre-training. The $R^2$ scores stay high on both the "covered" and "uncovered" subsets, indicating that the representation continues to identify the underlying state under a substantial change in the data‑generating mechanism.
> >
> > Regarding the reviewer’s concern that "consecutive observations are close" might be a trivial property of locomotion tasks:
> > **what our proof uses is not visual closeness in pixel space, but that conditioned on a skill $z$ the distribution of feature differences $\phi(o') - \phi(o)$** is sharply concentrated around $z$ on the hypersphere. This is a much stronger requirement than the temporal smoothness of the raw observations and is not guaranteed by the dynamics. In Appendix D.4, we test this property directly via G‑tests on the marginals of the learned feature differences. Apart from the HalfCheetah outlier mentioned above, these tests do not reveal systematic deviations from the high‑dimensional Gaussian/vMF behavior required by Assumption 2(ii), including in manipulation tasks where consecutive images can change substantially.

---

### Official Review · Reviewer_UPeg · 2025-11-01

**Soundness:** 3
**Presentation:** 2
**Contribution:** 3
**Rating:** 8
**Confidence:** 2

**Summary:**

Despite the popularity of mutual information skill learning (MISL) methods in RL, they are not theoretically well understood. This paper provides an explanation of why MISL methods work by drawing from the theory of non-linear independent component analysis (ICA). Specifically, the paper draws a connection between POMDPs and data generating processes in ICA, analyzes the feasibility of assumptions required for ICA results, and proves that the features learned by one particular MISL method, contrastive successor features, can be used to recover the underlying states of the (PO)MDP up to linear transformation. The paper also explains the utility of some algorithmic design choices from the ICA perspective. The paper further verifies the theoretical results by experiments in MuJoCo and Deep-Mind Control environments.

**Strengths:**

- The paper makes a novel connection between MISL methods and non-linear ICA.
- The paper proves identifiability results for CSF, providing insight into why the method succeeds and why certain design choices are better.
- The paper provides empirical results to back up the theoretical insights in a number of environments.
- The paper opens up a new direction of research for understanding self-supervised RL methods, and can be of wide interest to the RL community.

**Weaknesses:**

- The paper is dense and can sometimes be hard to follow.
	- For example, in Section 2.3, where the paper draws a connection between MISL and DGP, it was initially unclear to me how skills fit into the picture. It was mentioned in Section 2.1 that skills can be viewed as auxiliary variables, which can be brought up here again to aid explanation.
	- Perhaps due to a limitation in space, there's almost no spacing between some paragraphs.
	- There can be more discussion on the technical details of ICA in the background. This can make the paper more self-contained.
- The feasibility of some of the assumptions made by the authors to prove identifiability for CSF relies on empirical observations in a previous paper (Figure 2 of Zheng et al., 2024). However, it seems that Figure 2 of Zheng et al. (2024) presents results for METRA, not CSF.
- Line 478-480: "As exploration and state identification showed a positive correlation with the extrinsic oracle return, defined by each environment, this suggests that identifiability is helpful for zero-shot task transfer." But in Figure 3, there does not appear to be a correlation for the columns 3-5.
- References can be better polished. There exists papers without venue names, and some papers have the arXiv version cited when they were in fact published in conferences.

**Questions:**

- Do you foresee that some of the assumptions that rely on empirical observations (e.g., assumptions ii, iii, and iv) will hold for other MISL methods?

---

> ### Author Response · Authors · 2025-11-24
>
> We thank the reviewer for their acknowledgement of our work as providing novel insights into MISL methods and their detailed feedback, which improved the quality of our submission. We made the following changes to our submission (we detail our responses below):
>
> **Summary:**
> - We have restructured the background and related works and added more details on identifiability (Secs. 2 and 3)
> - Elaborated on our assumptions and Fig.2
> - We added more evaluations to demonstrate the advantage of an identifiable representation, i.e., generalization (Appx. D.6 and Fig. D.10)
>
>
>  ### Clarity
> Thank you for your suggestions on how to improve the clarity of our submission.
> We now explicitly mention in Sec. 2.3. that skills can be treated as the auxiliary variables in ICA and provide more technical details about the ICA framework (role of assumptions, general techniques)
> We clarify how we use Zheng et al. 2024 to motivate our assumptions
> We provide more experimental evaluation on identifiability and generalization
> We have replaced the references of preprints with published versions
>  ### Assumptions and Fig. 2
> > It  seems that Figure 2 of Zheng et al. (2024) presents results for METRA, not CSF.
>
> Thanks for pointing out that our formulation was unclear. Indeed, you are right, those results are about METRA.
> When we posit our assumptions for the identifiability proof of CSF, the important point is that the assumed data-generating process should be a (and not necessarily the only!) plausible description of the task. That is, if a representation satisfying those assumptions “works well” (and METRA, which is state-of-the-art on many tasks, does), then we can utilize those descriptors to posit our DGP.
>
> Furthermore, as Figures 3 and 4 of Zheng et al. (2024) demonstrated, METRA and CSF perform very similarly. However, we realize this nuance was missing from our original submission, which we now added to a footnote in the discussion of Assumption 1(ii) and also verified with CSF (cf. Figures D.5-8.), showing that Assm. 1(ii) holds almost universally in the investigated environments.
>
>
> > Do you foresee that some of the assumptions that rely on empirical observations (e.g., assumptions ii, iii, and iv) will hold for other MISL methods?
>
> That’s a great question! Yes, we do believe that these assumptions also hold for other MISL methods (DIAYN, VISR, LSD - for a comparison, see also Park et al., 2021), as many of them rely on the InfoNCE loss (ie., they share both the inner product parametrization, the representation learning aspect, and the cross-entropy-based objective), which is shown to be the most successful when these assumptions are fulfilled
>
>  ### Figure 3: reward and identifiability
> That’s a correct observation. We have corrected our statement in L478-480 and added the following context from the literature: “However, as it is well known in the identifiability literature, in some cases recovering more latent factors (i.e., better identifiability score) can be at odds with downstream performance on a particular task (Rusak et al., 2025)”
> That is, we emphasize that identifiability can be most useful when the downstream task(s) is (are) unknown---i.e., the compromise of an unsupervised approach is that the representation can be used for general tasks, but it might not excel at some of them, at least it might not reach the level of a methods that can exploit any form of supervision or prior knowledge.
>
> Furthermore, **to verify the premise of identifying the representation (i.e., generalization), we conduct further experiments (Appx. D.6 and Fig. D.10) to demonstrate that the representation learned by CSF identifies the ground-truth factors even if the trajectories were collected by another policy (so to speak, “offline”).**
> On the in-distribution trajectories (i.e., with states visited during training by our CSF agent), the $R^2$ scores show minimal degradation (in all four environments, they are around or above 0.9), whereas on the out-of-distribution trajectories (i.e., with not visited states), the $R^2$ scores are almost the same, except in Robobin, where they drop from around 0.9 to around 0.8, which is still considered a high score.
>
>  ### References
> Thank you for catching this! We have corrected the references.

---

### Author Response · Authors · 2025-11-27

Dear Reviewers,

Thank you again for your constructive feedback and suggestions to improve our submission. As the rebuttal period is coming to a close, we would like to kindly ask you to let us know if we have addressed all your questions and concerns, or if you have further questions.


Thank you in advance,
The authors

---

### Meta-Review · Area_Chair_4JHR · 2026-01-06

**Summary:**

This paper studies mutual information skill learning (MISL) in RL through the theoretical lens of ICA. Contrastive successor features (CSF) are used as the representation learning mechanism and proven to recover the ground truth environmental states up to a linear transformation. This result relies on specific design choices such as inner production parametrization and policy diversity. The authors validate these theoretical insights across various Mujoco and DeepMind control tasks, introduction a coverage dependent metric and demonstrating that the learned representations correlate strongly with ground-truth states.

In my opinion, this is a very interesting line of work to build an interactive map of the world and will be important as the community explores world models. The only other implicit map building RL approach in my understanding would be to build a successor representation (Dayan et al). This is sufficiently different and I would have liked to see this being discussed in the paper but this is an interesting line of work for the future.

**Reviewer Concerns:**

The reviewers initially raised several concerns, which were largely addressed during the rebuttal phase. The key ones are highlighted below:

1. Reviewers UPeg, vUpn, and YV3A questioned the realism of the theoretical assumptions required for the identifiability proof (e.g., equiprobable state differences, the mapping between consecutive states and skills). The authors clarified that the assumption regarding skill mapping does not imply a one-to-one correspondence between every transition and a skill, but rather that a discriminator can uniquely classify state pairs. They also provided empirical evidence (histograms) supporting the distributional assumptions (vMF/Gaussian behavior).

2. Reviewer YV3A and vUpn were concerned about the interpretation of high identifiability scores without sufficient baselines and questioned the correlation between identifiability and downstream performance. The authors clarified that the random network baseline corresponds to the initialization point in their plots. Authors also added significant new experiments during the rebuttal.

**Reviewer Scores:**

For the reviewers with the lowest scores:

vUpn: The rebuttal addressed the specific misunderstandings regarding definitions of skills/transitions, though the reviewer did not explicitly update the score.

YV3A: The reviewer actively engaged and acknowledged the value of the new generalization experiments and object-specific metrics.

---

### Decision · Program_Chairs · 2026-01-26

Accept (Poster)